# The Global Cross-Laminated Timber (CLT) Industry: A Systematic Review and a Sectoral Survey of Its Main Developers

**Victor De Araujo** [1,2,*] and **André Christoforo** [2,*]

1    Science and Engineering Institute, São Paulo State University, Itapeva 18409-010, Brazil
2    Exact Sciences and Technology Center, Federal University of São Carlos, São Carlos 13565-905, Brazil
*    Correspondence: engim.victor@yahoo.de (V.D.A.); christoforoal@yahoo.com.br (A.C.)

**Abstract:** Recently, both authors led a comprehensive review to discuss cross-laminated timber (CLT) as an engineered wood product, addressing the information and discussion on this building input in terms of the main details, materials, production forms, performances, codes, sustainability, applications, and perspectives for residential uses. The current scenario of CLT developers was raised in that previous paper as one of the missing factors in the available literature, and was the reason why this gap became the main goal of the present study. A global perspective was driven to provide information and discussion to every possible stakeholder. A systematic review on this sector was carried out, through the Web of Science and Scopus databases, to collect information and confirm this gap, using a representative method. CLT manufacturers were identified by their own websites, using a blended strategy formed by the systematic review results combined with the search for these companies using the Google search engine. Nearly a hundred CLT developers were identified and analyzed, in a representative sampling process. Results demonstrated that the CLT industry has manufacturing plants on the five habitable continents, which evinced a global commercial interest in this timber forest product. Despite the global presence of this mass timber product, most producers are concentrated in the northern hemisphere, despite being willing to serve international markets.

**Keywords:** wood products; building materials; timber buildings; wood industry; sectoral survey





## 1. Introduction

Looking at the category of engineered wood products for structural purposes, De Araujo et al. [1] remarked, in their comprehensive review, that cross-laminated timber is processed into industrialized elements used in the modularization of partial-to-finished modules and prefabrication of floors, roofs, walls and short walls, and staircases. A light weight is an important characteristic of this multi-layer panel [2]. Cross-laminated timber (CLT) is able to present a better fire behavior in load-bearing walls than structural slabs, being influenced by the thickness and number of layers [3]. Formed by an uneven number of orthogonal layers (lamellae), this highly industrialized solution is a glued and pressed panelized product with variable dimensions, which include widths between 50 cm to 3 m, thicknesses ranging from 30 to 40 cm, and lengths that can exceed 18 m and, occasionally, reach 30 m for some specific uses [4–8].

Glued-laminated timber construction offers a more sustainable alternative to precast reinforced concrete, due to reduced emissions and increased carbon dioxide sequestration [9]. CLT-based buildings can produce 27% less of global warming potential emissions than concrete buildings, insofar as this engineered wood panel could significantly reduce the embodied energy by 40%, due to the use of solid wood [10,11].

Featuring convenient sustainable and structural advantages, CLT products have been increasingly used in industrialized buildings [1]. This was predicted by the FAO/UN [12], as the global production capacity was about three million cubic meters of cross-laminated timber in 2020, with the possibility of an increase in the short term.

In the field of timber construction, its industrialized products vary from simplified and semi-finished examples dependent on additional processing to finished modules, of the turnkey type, ready for on-site assembly [13]. In the industrialized construction systems, the transfer process of production functions and resources is developed towards a more stationary industry for a production based on decentralized systems [13,14]. In practice, an industrialized building is substantially developed in manufacturing plants, insofar as prefabricated parts are shipped in kits or (semi)finished modules to multiple or single construction sites.

Intense plant prefabrication may cause substantial impacts on the larger-scale production performance of timber construction systems, as it allows for the achievement of efficient levels of repeatability and reliability [13]. Higher prefabrication naturally demands industry plants aimed at manufacturing an intensively engineered product in a production line, provided with openings and spaces for frames and built-in installations, suitable for quick fixations and assemblies on the construction site [2,15]. As for fixation, component connections may require nails, dowels, screws, and other metal connectors, galvanized or aluminized, which allows more efficient behaviors regarding lateral loads [8,16,17]. In the stage of assembly, all engineered wood products (e.g., cross-laminated timber panels, glued-laminated timber beams, etc.) are transported by trucks, from the manufacturing plants to the ground where the construction site is located so that the inputs are moved by cranes to the installation and/or fusion positions.

The variety of industrialized options on the market is complex and extensive, as many producers have already sold their own construction systems developed from their engineered wood products to satisfy clients, needs, and limitations [13]. These solutions usually blend architectural concepts, complex styles and diversified raw materials, since a modern solution can be prefabricated on a large scale, either in standardized or custom form, taking advantage of distinct wood strengths, densities, textures, shapes and colors.

This multiplicity is evident even in the product choice. For example, cross-laminated timber (CLT) is about 7% more expensive than glued-laminated timber (GLT) [11]. This economic difference can be easily reversed. While GLT beams are interconnected to form a structural frame and require additional materials to close walls and floors, CLT panels already act as shearing walls and floors. Therefore, this simplicity becomes an advantage of CLT products in the manufacture of industrialized buildings.

After almost three decades since the creation of cross-laminated timber, Muszynski et al. [18] identified the fact that its industry is young and unknown in many emerging markets, especially in the case of wood utilization for construction, which is a reason why this production sector is globally promising and can gain new entrants and stakeholders.

The novelty of this topic is justified by the low number of studies on this industry. Led by the authors of this present paper, their research group also developed a complete review to address and discuss cross-laminated timber as an engineered wood product for industrialized buildings [1], and their contribution confirmed this lack of information about the CLT industry. Due to this gap previously identified by the authors in their review, the present study aimed to evaluate its coverage and global presence, and analyze CLT producers by observing their official websites to provide information and stimulate new synergies and industry collaborations. As a result, three main issues were observed:

- There are few CLT producers, and they are limited to Europe and North America;
- These manufacturers do not value product advantages and environmental benefits provided by the use of this modern bioproduct in construction;
- Most CLT developers still prioritize their national markets.

## 2. Materials and Methods

*2.1. Strategy of Two-Stage Search: Systematic Review and Sectoral Survey*

The characterization study of the global cross-laminated timber industry was developed from two methodological procedures. The first stage included a systematic review

in order to identify and collect information from scientific papers and technical standards and codes.

The second stage was represented by a sectoral study applied to producers of this massive construction input in order to understand representativeness, corporate goals, geographical coverage, solutions, markets, environmental benefits, and other contents.

This double methodology prioritizes the identification of the current moment of this global industry by comparing information from publications collected in the first stage with new findings identified by the second stage, using surveys of official websites of the producers of cross-laminated timber (CLT). The main steps for each methodology are detailed in the following sections.

### 2.2. Systematic Review: Searches and Codifications of the Search Process

This process used a multi-level search, as two scientific databases were utilized to identify recent information and discuss the literary scenario of the global CLT industry.

Due to the global objective of this study, the literature searches only prioritized publications written in the English language, without any restriction regarding the period of publication, in order to consider all existing publications on this topic.

Like the primary research review on CLT products [1], this study also verified the representativeness of the cross-laminated timber industry through two scientific databases, Web of Science and Scopus, using keywords described in Table 1 with the Boolean operators "AND" and "OR". Two conditions of searching were utilized to intensify the search, including a general condition formed by individual terms (product and corporate types), and a specific topic about the main goal (CLT industry) with individual terms of the product.

**Table 1.** Conditions and respective keywords.

| Condition | Strings |
|---|---|
| General | (("cross-laminated timber" OR "cross laminated timber") AND ("industry" OR "industrialization" OR "producer" OR "developer")) |
| Specific | (("cross-laminated timber" OR "cross laminated timber") AND ("CLT industry")) |

### 2.3. Systematic Review: Synthesis and Results Presentation of the Search Process

From both conditions raised by Table 1, their search results were recorded on spreadsheets of the Microsoft Excel 2016® package (version 16.0, Microsoft Corporation, Redmond, WA, USA), whose bibliometric data from databases was duly extracted in compatible formats (XLS and CSV extensions), using the same strategy as that led by De Araujo et al. [1] to quantify publication metrics in graphs and tables such as authorships, types, origins, years, and volumes per year. Due to high representativeness, the demonstration of results only considered those authors with three or more publications in any database. As in the first research, the "conference papers" and "proceeding papers" were considered equivalent, that is, they are part of the same publication category.

This projection was carried out through the insertion of bibliometric data organized in specific spreadsheets and, therefore, saved for a research record. The analysis using two different conditions, a more general and another more specific (Table 1), was possible in both scientific databases, which enabled a triangulated analysis through two different data sources, with global coverage.

### 2.4. Sectoral Survey: Searches and Codifications of the Search Process

Although there are few manufacturers focused on the production of a product made of wooden lamellas fixed by connecting elements (screws, pins and plugs) as verified by [19], the present study only prioritized the CLT industry dedicated to glued products.

In the context of the global perspective of the CLT industry, the identification of its producers was developed from searches using three different search engines: Scopus and Web of Science databases, and Google® web searcher.

Within the scope of both scientific databases, papers published in peer-reviewed journals, papers of conference proceedings, books and book chapters searched for in the previous stage given by the systematic review were used to identify potential developers. Aiming at a representative and robust analysis, this search process considered these different information sources in order to formalize a list of existing CLT producers. During the review process, this list was updated in line with the reviewers' recommendations.

Alternatively, the systematization through the Google® search engine used searches carried out from the typing of individual terms (Table 2), which is the reason why Boolean operators were not utilized. Standards and codes, reports, technical manuals and catalogues were identified and saved during the reading stage of the results for each individual search. Simultaneously, some news and reports were also read and saved when they revealed any potential manufacturer's name.

**Table 2.** Keywords used to search for CLT producers.

| Condition | Strings |
|---|---|
| General | "cross-laminated timber", "cross laminated timber", "Xlam", "X-lam", and "CLT" |
| Specific | "CLT production", "CLT industry", "CLT manufacturer", and "CLT producer" |

During the searches in the sources of information considered, the name of each identified producer was noted, as well as the municipalities and countries that host their factories (industry plants), in electronic spreadsheets in Excel. In order to confirm the existence of these potential developers, the name of each company was individually searched by Google® and LinkedIn® to identify official representation, either by its social profile or its website. This stage was carried out from November 2022 to March 2023.

The identification of the condition of all manufacturers and the geographic locations of their factories depended on valid access to the websites. This step allowed this verification of the compliance of the activities of the companies previously identified. This verification by official representations was efficiently carried out by [20], insofar as searches for corporate profiles on Instagram® allowed the identification and confirmation of the existence of timber construction companies in Brazil, and provided a representative characterization of this national industry regarding the corporate possibilities and perspectives with respect to their activities with e-commerce.

### 2.5. Sectoral Survey: Development of the Structured Script to Collect Data Using Websites

From the list of CLT manufacturers, the second stage of the sectoral study was carried out through the utilization of their official websites, in order to analyze the corporate data and, therefore, characterize this global industry through a representative sample.

Only information shared in the official websites was considered in this investigation, ensuring a more assertive scenario. From contents disclosed by companies, five questions were investigated, in order to understand the corporate aspects (Table 3). This analysis still allowed for the characterization of the global CLT industry, collecting data anonymously and carefully, on the geographical location of manufacturing plants currently in operation, the corporative goals of this population, product details, and customer contact channels.

**Table 3.** Questions and aspects under evaluation in this exploratory survey.

| Question | Justification | Alternative |
|---|---|---|
| Query 1: identify the company's location | Understand if companies specify their locations for customers | Country; state; city; headquarters address; factory address |
| Query 2: identify the customer contact channels | Understand the channels shared by companies to establish formal communication with customers | Message box; wireline phone; mobile phone; fax; WhatsApp; e-mail; Youtube®; Facebook®; Twitter®; Xing®; Instagram®; VK®; LinkedIn® |
| Query 3: identify all communication languages | Understand the company's access to the foreign market by delivering content in different languages | Free alternatives [1] |
| Query 4: identify corporate structures and attributes with respect to data and values | Understand if companies clearly expose their goals, conditions and values that form the ambitions and structures of corporations | History; activities/businesses; vision/goals; news; work teams; production scheme; innovation; technical certification; forest management |
| Query 5: identify all products/services | Understand the market objectives and product details from shared information | Product description; product photos; product performance; product sustainability; product advantages; compact building works; large building works |

[1.] languages identified by research manager from different information (designations or flags).

The methodology applied in this stage was based on an adaptation of some issues studied by [20] in a sectoral survey of a domestic industry perspective. Unlike this nationally oriented approach, the global coverage of the present sectoral survey allowed for the delineation of a significant analysis. The observation of corporate data, either of textual or visual contents, supported the characterization of population scenarios for each issue listed in Table 3. The specificities and conditions which compose this analysis of the global CLT industry are met by identifying the presence or absence of each alternative.

The survey was initially carried out in January 2023. Due to a review process, the list of developers and the sectoral evaluation were updated, and concluded in March 2023.

### 2.6. Sectoral Survey: Synthesis and Results Presentation of the Search Process

Each CLT producer was analyzed from the information shared on its official website. Corporate data were collected and organized according to the five queries set out in Table 3. The individual results served to build a global perspective of the current condition of this industry. Thus, data was recorded in spreadsheets using Excel to produce sectoral graphs and tables to measure each alternative observed for each question and, therefore, to compare the popularity of each item.

As this survey was based on a sampling process, the margin of error (E) was utilized to calculate the research error, using the statistical software developed by Raosoft [21]. The calculation considered two prescriptions of this program developer, which included a 95% confidence level and a 50% response distribution.

### 2.7. Discussions: Synthesis and Presentation

In view of the conclusion of the bibliometric results, each searched document was read individually in the case of its title and abstract, in order to categorize its objectives and confirm those studies essentially focused on the CLT industry and its developers.

Different publications were selected to compose the discussion topic, among which documents not included in the two databases were also used, including reports, technical codes, conference presentations, and reports published in the media representing a general and a niche scope. This strategy was considered due to the specificity of the industry topic, as this sector is exclusively oriented to a type of engineered wood product in use for construction. It was possible to develop different approaches for an unprecedented sectoral survey in order to identify each CLT producer and forecast the current scenarios of this industry by examining different issues and using a representative sampling.

## 3. Results and Discussion

### 3.1. Results of Systematic Review

The bibliometric analysis revealed the representativeness of scientific publications, both from a global perspective (Table 4) and in their distribution over time (Figure 1), for two refinement conditions and using two databases for early 2023, as specified in Table 1.

**Table 4.** Volumes of prospected documents according to different contexts and sources.

| Condition | Scopus | Web of Science |
|---|---|---|
| General | 197 | 112 |
| Specific | 4 | 2 |

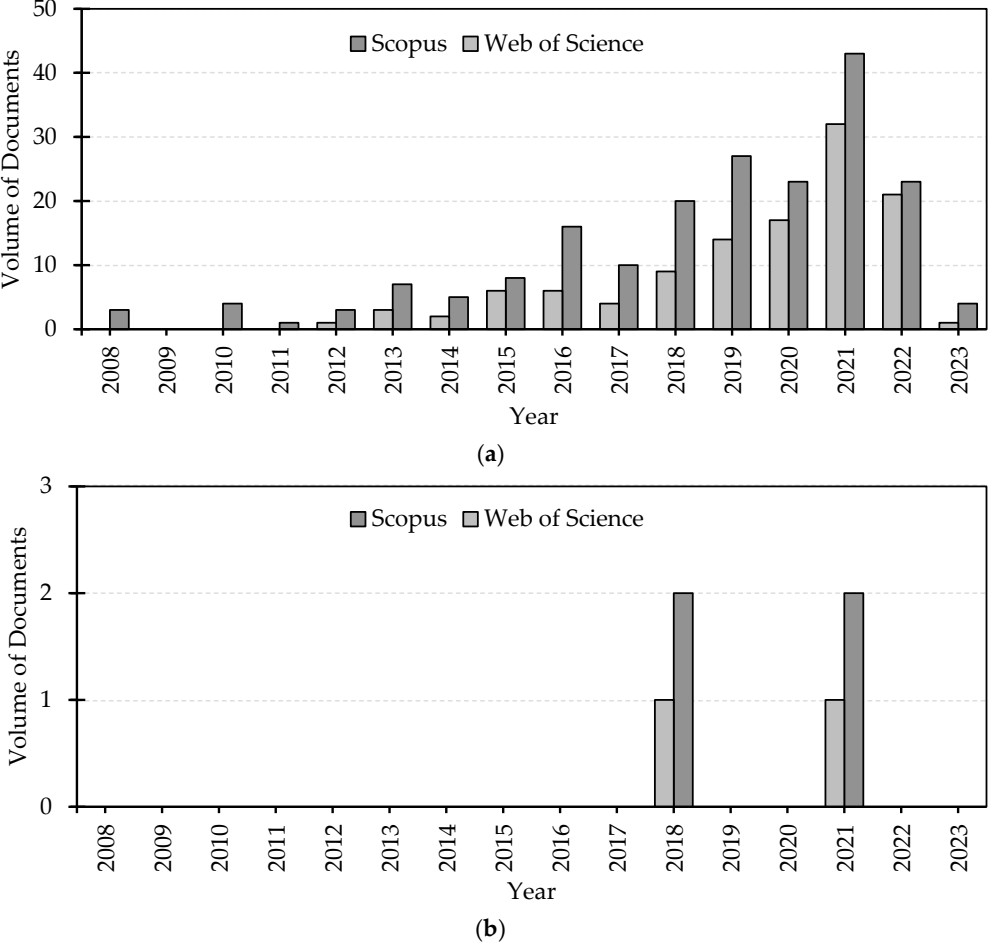

**(a)**

**(b)**

**Figure 1.** Annual volumes of publications related to the CLT industry by database under observed conditions: (**a**) general and (**b**) specific.

The volume of documents searched for using the Scopus database in the general condition exceeded 57% of the publications available in the Web of Science, being 50% higher than in the specific condition verified by Scopus (Table 4). Despite the growing number of publications about industrial contexts related to cross-laminated timber products (Figure 1a), the scenario of a specific condition is still perceptibly small (Figure 1b). Year after year, the Scopus advantage was also repeated under the observed conditions (Figure 1a,b). This greater volume of publications obtained by the Scopus database was already expected, due to the greater number of indexed scientific journals when this scientific platform is compared to the Web of Science, as verified by Pranckuté [22].

In the general condition, a hundred scientific documents are currently available on the Web of Science, although twice that amount is already listed in Scopus (Table 4). In this observed condition, 110 journal articles, 80 conference articles, 6 book chapters, and an editorial were identified in the Scopus database.

Two hundred authors were confirmed in the Web of Science, and 150 authors were identified in Scopus. Due to these expressive amounts, Figure 2 detailed only those authors with three or more publications in at least one of the analyzed databases.

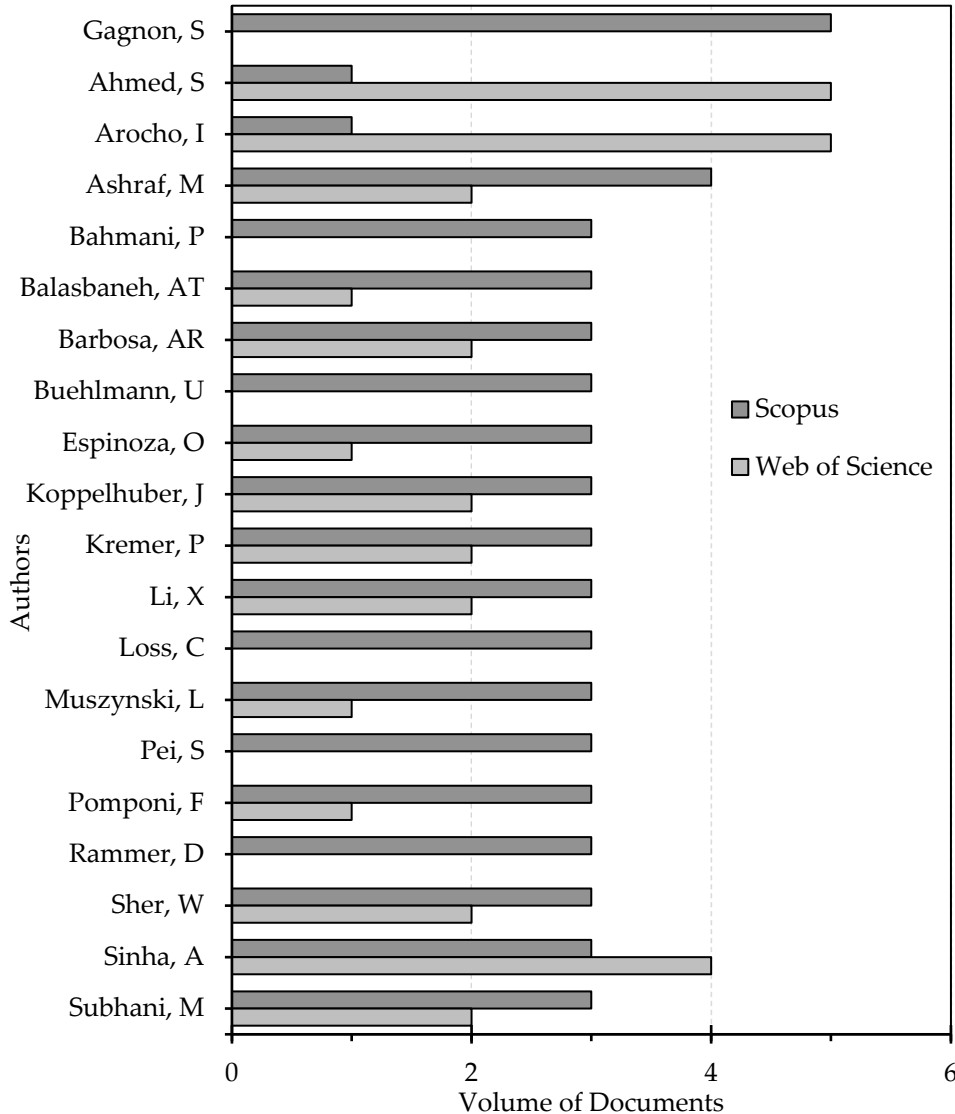

**Figure 2.** Volume of publications per author related to the CLT industry under the general level by database.

Thus, the ranking of the main contributors specified 20 academics that satisfy this criterion in the year 2023. Sylvain Gagnon, Shafayet Ahmed and Ingrid Arocho were among the most active authors (Figure 2) who carry out studies in Canada and the United States. In addition to these North American countries (Figure 3a), Australia, the United Kingdom, Austria and Sweden are among the countries with the highest number of publications in the general condition.

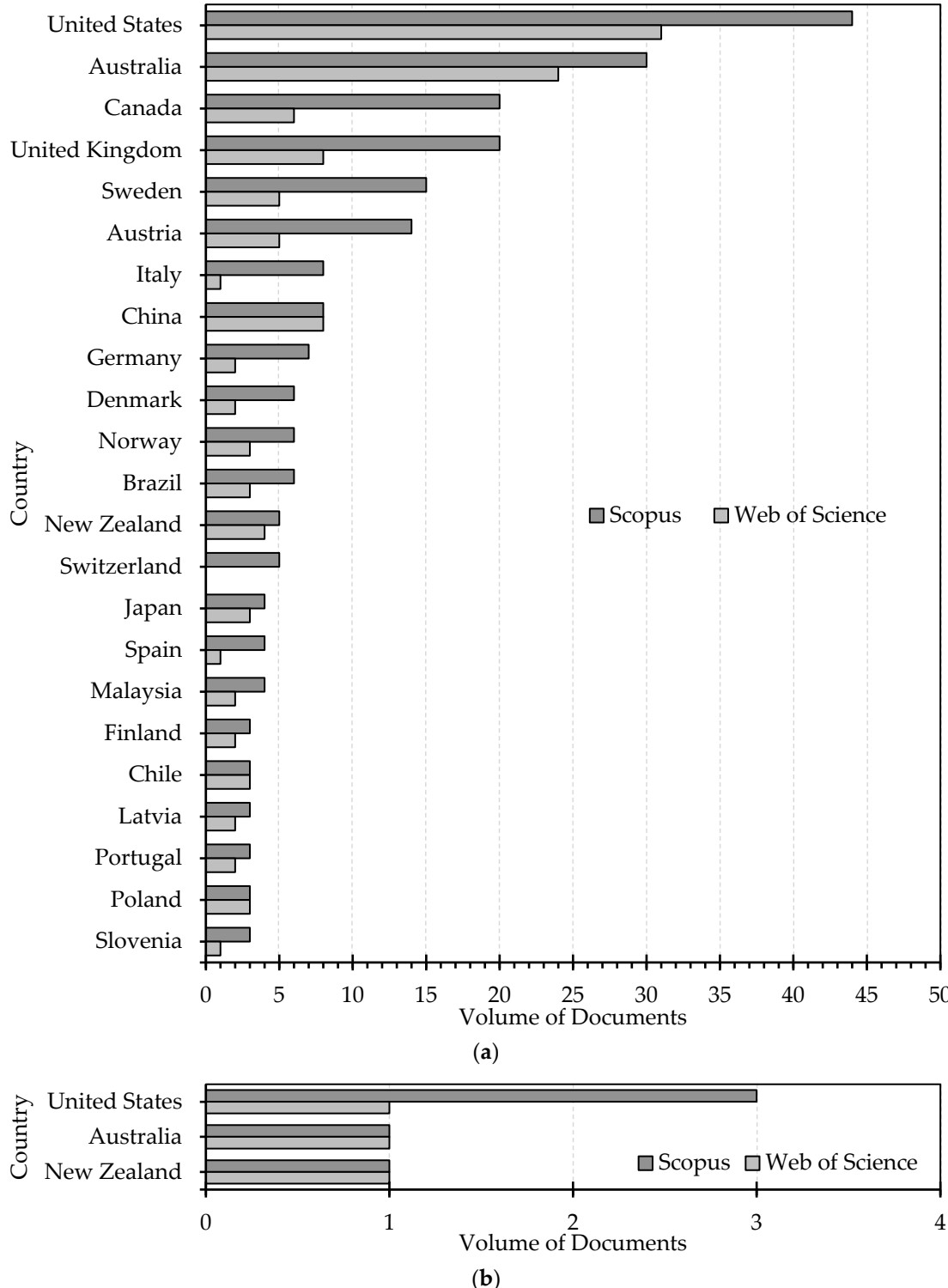

**Figure 3.** Volume of publications per region related to the CLT industry by database, under observed conditions: (**a**) general and (**b**) specific.

Authors from Latin American nations, such as Brazil and Chile, are also present in publications contained in the general condition, which is more comprehensive (Figure 3a). Due to a higher specificity, the specific condition has only four documents (Figure 3b), which are from Oceanian and American nations.

In view of the searched-for results to compose the arguments of this study (Table 4), the reading of the title and abstract of each publication allowed for the categorization of its main goals and fields of study. The low number of scientific publications aimed at the study of the cross-laminated timber industry was confirmed (Figure 1b).

Due to these few documents identified in a specific level and the greater volume of documents indexed by Scopus, the categorization of the main objectives was applied to those publications available in Scopus for the general level (Figure 4). Thereby, it was possible to verify that most of the documents on cross-laminated timber of this more comprehensive type, despite the industrial context, were related to construction uses within a structural, environmental, seismic, design, production, technical, and economic analysis.

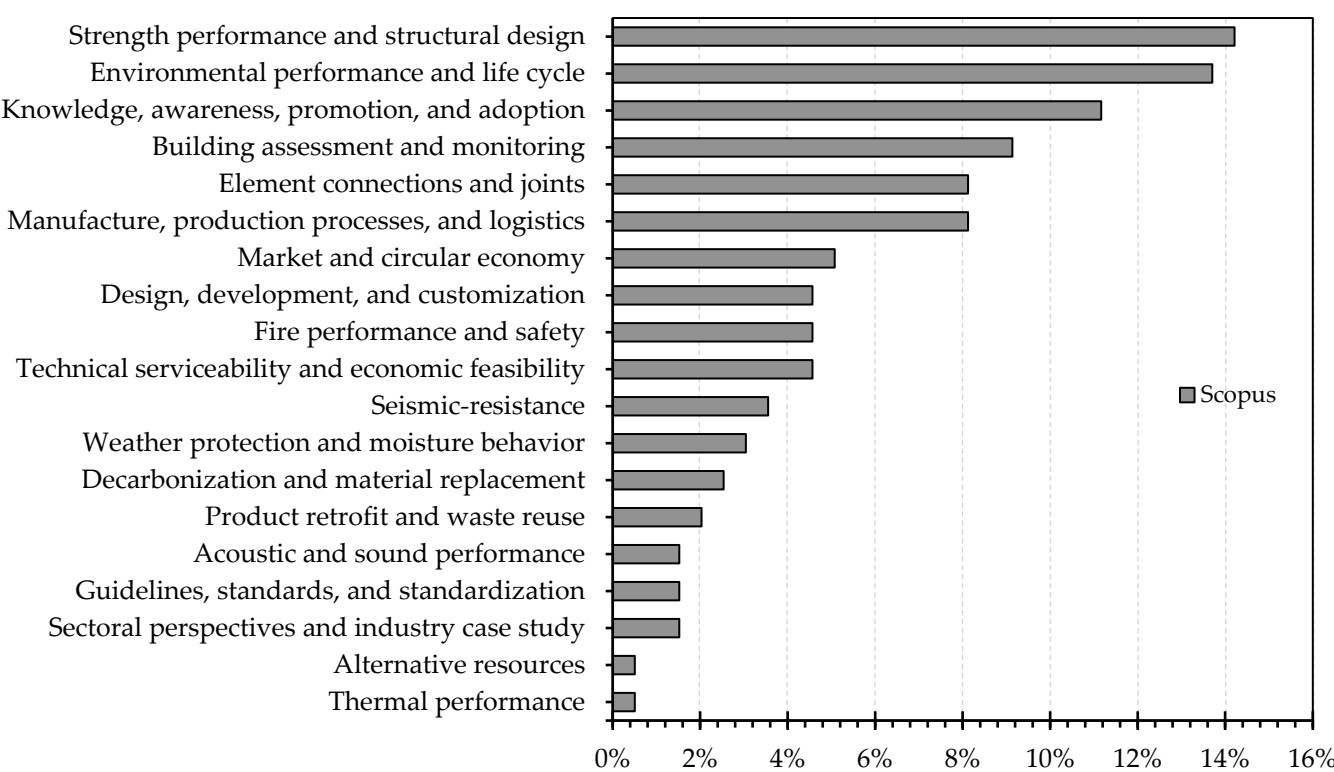

**Figure 4.** Popularities of main goals addressed by publications searched for in the general level according to Scopus database.

Among those fields addressed in the general condition, three studies were identified (Figure 4), but only one publication addressed the global cross-laminated timber industry. In this single contribution, Larasatie et al. [23] identified 66 CLT producers, of which only 12 companies were duly studied, using an interview process driven by a formal questionnaire.

*3.2. Results of Sectoral Survey*

This section is organized according to the identification of cross-laminated timber developers and location of each factory dedicated to the industrial production of glued CLT products, and the characterization of this global industry based on sampling and questioning.

3.2.1. Identification of Developers and Their Geographic Locations

Through the systematic review, a single study addressed the cross-laminated timber industry, which was led by Larasatie et al. [23]. In contrast, further searches carried out by

the research manager, using Google search engine and publications cited by other studies, also identified four other documents about the CLT industry, which included a journal paper developed by Muszynski et al. [24], two conference papers by Muszynski et al. [18,25], and a workshop presentation for wood panels presented by Muszynski et al. [26]. Both contributions were led and expounded by the same research leader.

While 19 companies and their locations were revealed by [26] in 2016, 21 companies were surveyed without information of their designations and locations [24], and the recent estimates led by [18,23] realized that this global industry could be up to four times larger than the initial total of [26].

Given the lack of corporate information and this low number of studies, the scenario reinforces the need for a complete survey to understand this global industry. Thereby, these reasons justified the development of the present study, as the same was designed to identify each CLT developer by its corporate name, discover the manufacturing coverage around the world, and characterize its main aspects and goals.

Therefore, the steps of identification and confirmation of CLT manufacturers were the starting point of this sectoral survey, making its development possible with regard to the representative characterization of this global industry. The elaboration of the sectoral list began with those 19 producers exemplified by Muszynski et al. [26], and this total was expanded with the addition of other manufacturers specified by some reports, magazines and technical codes identified in the searches using the Google® search engine (Table 2); these further sources were given by [19,27–34]. The list was completed by the identification of CLT manufacturers, whose official websites were randomly discovered during this prospective stage.

The process to certify the operational activity of each manufacturer was carried out by nominal searches of each company name recorded in the listing developed in this step. Thus, it was possible to confirm the official website and social profile using Google® and LinkedIn® search engines. Thus, this complete list disclosed the names and locations of each confirmed company, including cities and regions where each CLT manufacturing plant is currently located; see Appendix A for the detailed designations.

The sectoral list specified 98 manufacturers of cross-laminated timber (Appendix A) in March 2023. This quantification exposed a more numerous global industry—which is 33% larger in terms of company number compared to the sector estimated by [18,23,25].

Globally, CLT developers are already present in the five habitable continents, that is, Africa, America, Asia, Europe, and Oceania. However, most of this global industry is located in the European and American continents (Tables A1 and A2), as both regions contain 70 and 18 producers, respectively. Eight companies are already active in Asia, while one company was identified in Oceania and another in Africa (Tables A3–A5).

In another analysis, the predominance of concentrations becomes greater when comparing the business concentration according to the hemisphere, that is, through the observation of territories with CLT developers located above and below the equator. While 92 of the developers are located in countries in the northern hemisphere, only six producers include territories in the southern hemisphere (Figure 5). Due to its continental dimensions, Brazil is the only southern territory contained in the transition region of both hemispheres. No other country is geographically close to this line of latitude.

Europe represents the continent where the largest number of cross-laminated timber manufacturers was identified. There, 70 producers are in 18 distinct countries (Appendix A, Table A2). The Americas present the second largest continental population, where 18 CLT producers are distributed in five countries, including two nations in the northern subcontinent and another three nations in South America (Appendix A, Table A1). Asia already includes eight CLT producers, all located in Japan (Table A3). A CLT producer is verified in Africa (Table A4), while another company originates from Oceania (Table A5).

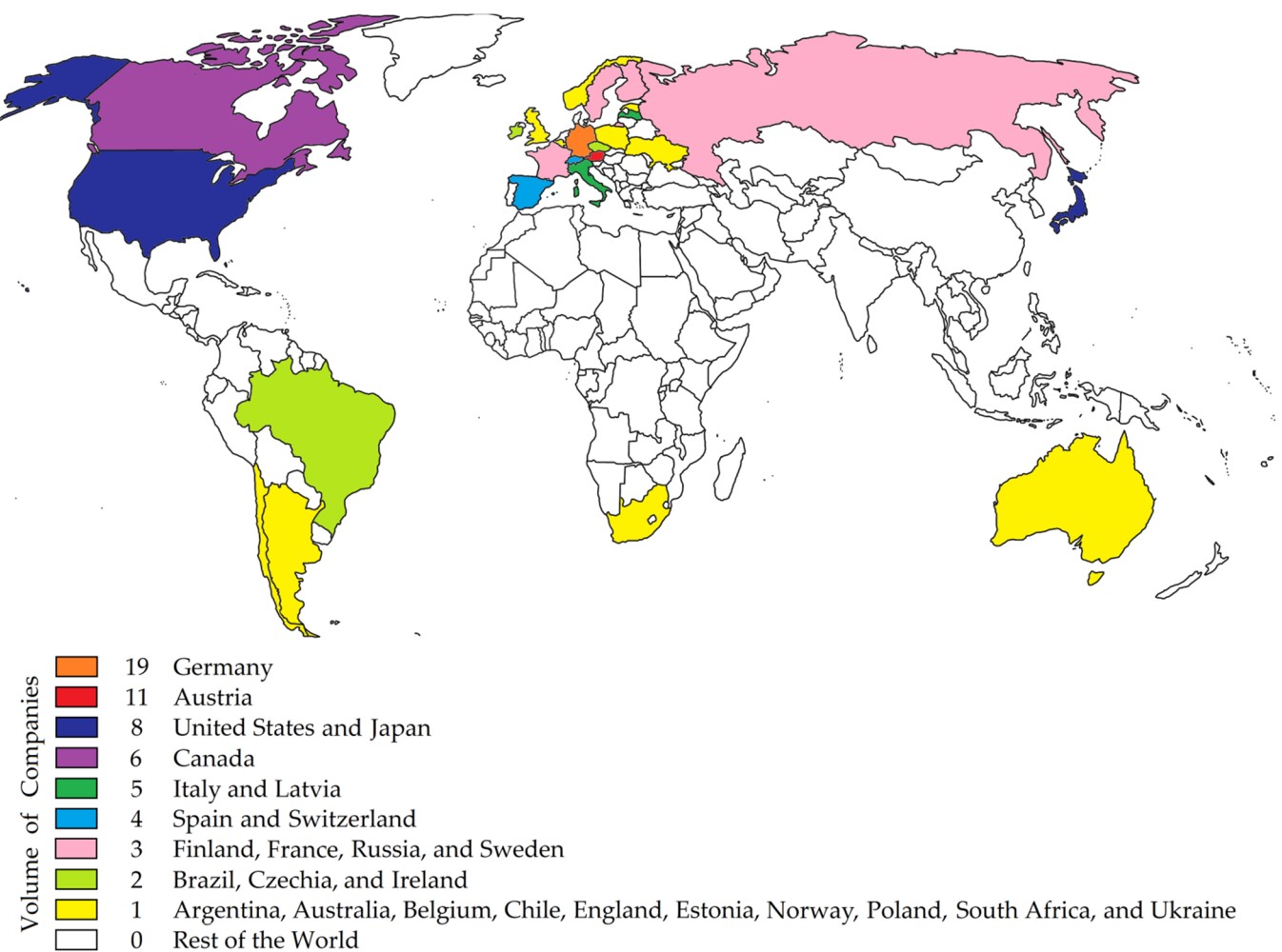

Volume of Companies

| | | |
|---|---|---|
| 🟧 | 19 | Germany |
| 🟥 | 11 | Austria |
| 🟦 | 8 | United States and Japan |
| 🟪 | 6 | Canada |
| 🟩 | 5 | Italy and Latvia |
| 🟦 | 4 | Spain and Switzerland |
| 🩷 | 3 | Finland, France, Russia, and Sweden |
| 🟩 | 2 | Brazil, Czechia, and Ireland |
| 🟨 | 1 | Argentina, Australia, Belgium, Chile, England, Estonia, Norway, Poland, South Africa, and Ukraine |
| ⬜ | 0 | Rest of the World |

**Figure 5.** Distribution of active CLT manufacturers according their origins in early 2023.

The stage of identification of CLT developers and the respective geographic locations involved a diversified perspective, in which the obtained results demonstrate the effective global presence of this industry. Figure 5 shows the distribution of each CLT developer, according to the origin represented by its headquarters location in early 2023, that is, by its main home country. In practice, this finding becomes extremely important both for the economy of the wood industry and the modernization of construction, insofar as there is a very representative number of active developers aiming to supply the construction activities through this modern engineered wood product.

3.2.2. Characterization of the Global Industry of Cross-Laminated Timber: Sampling

Through the formal definition of the observed population, which is nominally and geographically identified in Appendix A according to each individual searched for in the previous stage, it was possible to proceed with this sectoral survey to characterize and discuss the global industry aimed at glued products of cross-laminated timber in 2023.

Despite the consideration of this complete listing exemplified in Appendix A, only manufacturers with official websites could be evaluated in complete compliance with the expectations and requirements considered in the methodology of this survey. The list of companies and the sectoral survey were updated in the review stage in March 2023. Only one manufacturer, located in Europe, was not included in this sampling process, as it does not provide an official website and, therefore, this company does not meet the requirements of this analysis. Thereby, this investigation analyzed the global industry of cross-laminated

timber in almost all of its population oriented towards glued solutions. Table 5 described the results of the survey sampling applied to the global population.

**Table 5.** Global sector and survey sample in March 2023.

| Population in 2023 | Company Volume (Unit) | Sectoral Percentage (%) | Margin of Error (%) |
|---|---|---|---|
| Identified developers | 98 | 100 | – |
| Sampled developers | 97 * | 98.98 | 1.01 |

* one company was not sampled because it lacks an official means of observation.

Both in percentage terms of the number of companies sampled and the margin of error obtained, high levels were achieved as a result of sampling (Table 5). This outcome shows a very representative analysis, as 1% of the total population was not surveyed.

The obtained margin of error (E = 1.01%—that is, ±0.51%) indicates a highly reliable scenario, since Pinheiro et al. [35] prescribed, as an ideal condition, a sampling with 0% and ±2.5% of errors. As a result, it was possible to obtain an unprecedented and very representative sectoral evaluation of the global cross-laminated timber industry.

This survey focused on different issues, detailed in Table 3. These five questions were designed to facilitate the evaluation of each corporate site and enable the study of the variables under observation in each issue. Different contexts were analyzed to formally represent the current panorama of this industry through information disclosed by these developers on their websites. This second stage omitted the identifications of all sampled companies, grouping the total results per variable, in order to provide a characterization of the global industry rather than an individualized analysis of companies.

3.2.3. Characterization of the Global Industry of Cross-Laminated Timber: Questioning

The first question analyzed the degree of specification on the websites in sharing the geographic location(s) of the facilities (headquarters and factories). To avoid duplication of data, each CLT developer with more than a single factory was counted only once, in its country of origin. From 97 sampled producers (Table 5), only 5% (E = ±0.51%) of these individuals did not detail their locations to customers through their official websites. In contrast, 92% of this analyzed population specified both cities and full addresses of headquarters (Figure 6a). However, 53% of this industry gave complete details of the name of the street(s) where the factory(s) was located, while 14 companies had two or more manufacturing facilities, as specified in Appendix A. In turn, 41% of companies shared their regions as designated by the federal state of their possible locations, although only 52% of this same global sector prioritized the specification of their own nations (Figure 6a). It should be noted that locations are not always clearly presented (Figure 6b), since an additional observation confirmed that 52% of this industry already shares the corporate location at some point on the home page of the evaluated websites, which requires users to look for this information in the items or tabs referring to contacts, locations and histories, being used by 71%, 15% and 5% of the producers, respectively.

A noticeable part of this global industry still needs to specify the complete location in order to elucidate the geographical presence for (current and potential) clients, partners, and suppliers (Figure 6a,b).

The second question dealt with the communication channels available on websites (Figure 7). This observation revealed that companies have prioritized the declaration of their wireline phones and e-mails, because they are tools used by 93% and 84% of this industry, respectively. Direct contact with customers established through message boxes inserted on these websites was an alternative offered by 68% of this studied sample. The communication by mobile phone and WhatsApp® was shared by 12% and 4% of these official websites. Due to the massive popularization of smartphones worldwide, a more expressive sharing of mobile numbers was expected, because WhatsApp® requires this

numerical information to establish contacts on different devices. Despite the status of disuse, fax numbers still represented an alternative available to establish communication, as this information was present on 30% of the websites (E = ±0.51%).

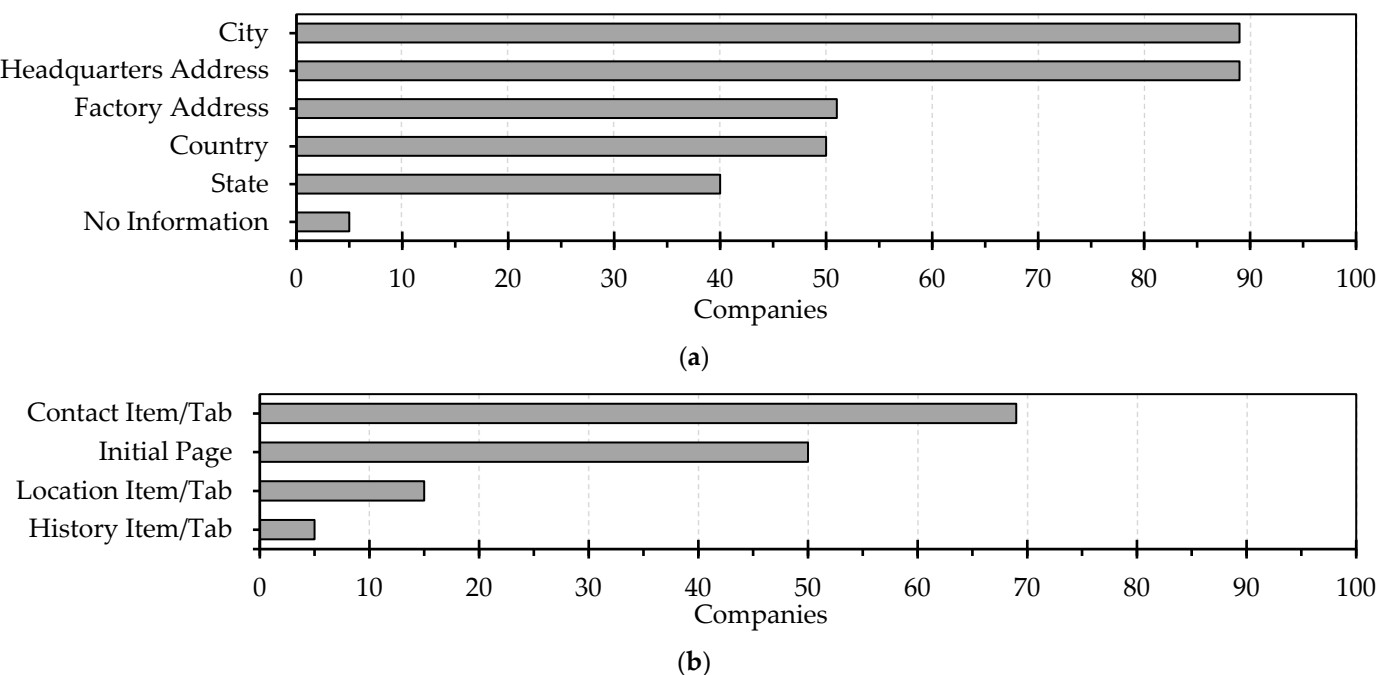

(**a**)

(**b**)

**Figure 6.** First question: locations of CLT developers regarding (**a**) the level of specification shared with customers, and (**b**) where this information is available on the website. (n = 97; t = March 2023).

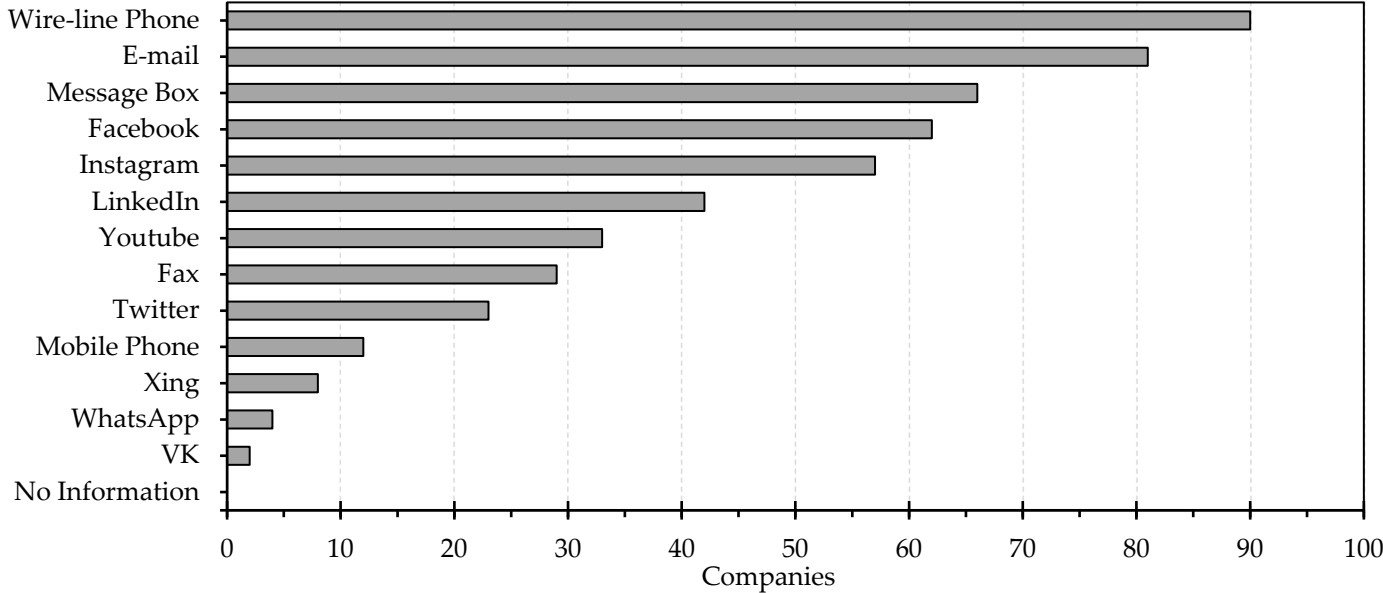

**Figure 7.** Second question: customer contact channels available on the websites of CLT developers. (n = 97; t = March 2023).

Social networks have been essential resources to be used by any sector, which is the reason why Facebook®, Instagram®, LinkedIn®, YouTube®, Twitter®, Xing® and VK® became the profiles for communication offered by 64%, 59%, 43%, 34%, 24%, 8% and 2% of the producers, respectively (Figure 7). These profiles may assist the company in the

mass dissemination, free of charge, to different audiences, of any type of audiovisual media and content.

In the third question on the communication representation, 24 different languages were identified as formal dialects available on the evaluated websites (Figure 8a).

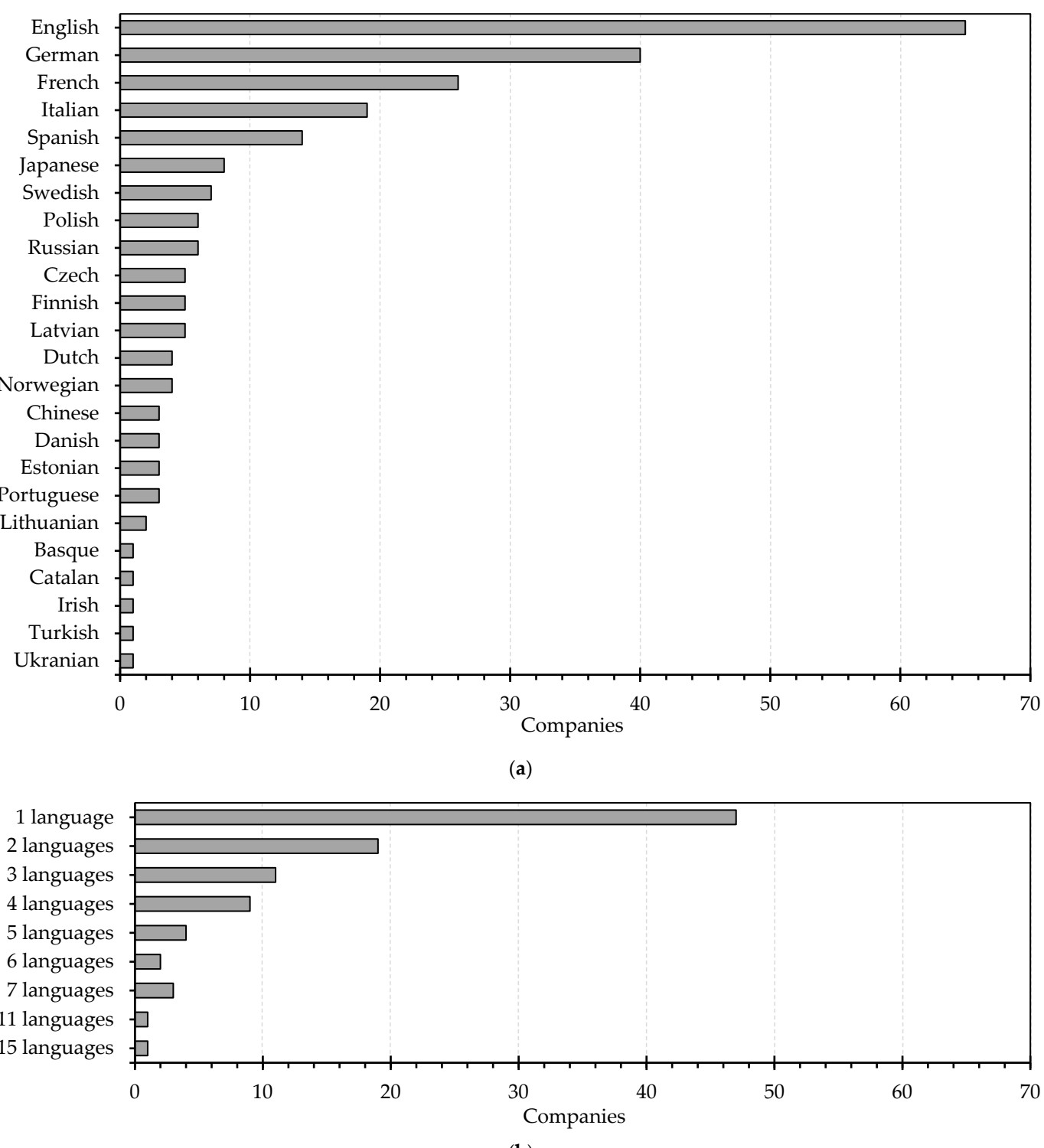

(**a**)

(**b**)

**Figure 8.** Third question: languages shared by official websites of CLT developers regarding (**a**) type of language, and (**b**) number of languages offered by company. (n = 97; t = March 2023).

The main option offered by companies was the English language, being present in 67% of the websites of sampled developers. German, French, Italian and Spanish represented other very popular languages, which were available on 41%, 27%, 20% and 14% of these official websites (E = ±0.51%), respectively (Figure 8a). Other 19 different languages did not individually reach a presence of 10% in this studied population, although Japanese and Swedish were available on 8% and 7% of websites, respectively.

Regional languages were identified as well. While an official website was offered in Catalan, Basque was used in another website (Figure 8a). Although these two companies also shared Spanish as an alternative language, Catalan and Basque were available due to regional issues, as these dialects are commonly used by native people from the Catalonia and Basque Country—they are autonomous communities of Spain.

An observation about language context included another perspective, which was represented by the number of languages made available on the websites (Figure 8b). Two CLT manufacturers exceeded the availability of 10 or more languages on their websites, this being a very favorable condition to attract a greater number of populations from different origins and territories. A broad availability of languages was identified in 5% of the sector; 3% of companies used seven languages and 2% offered six languages on their websites. In 4% of producers, five languages were available on their websites. Around 9% of the population offered four languages, and trilingual websites were verified in other 11% of the same. Bilingual websites were confirmed in 20% of this studied sample (Figure 8b). In contrast, 47 manufacturers (48%, being E = ±0.51%) exposed their products and services using only a single language (Figure 8b), that is, these were websites using the native languages of their territories. It should be mentioned that the English language was utilized as the single form of communication in 19% of this sampled population.

Although the declaration of the companies' activities and businesses is fundamental information, as it defines corporate functions for the market, this content was still absent in 2% of the producers in the fourth question about corporate goals and attributes (Figure 9). The presentation of corporate objectives and visions on websites, as a means to clarify corporate expectations and targets, was unavailable for 4% of this population.

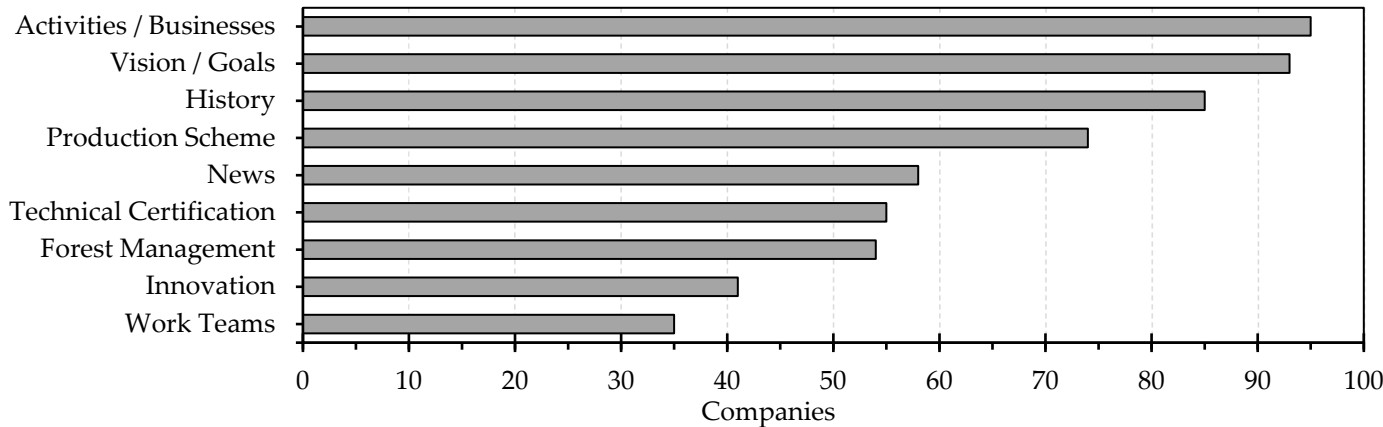

**Figure 9.** Fourth question: goals and attributes shared on the websites of CLT developers. (n = 97; t = March 2023).

The history of any company aims to contextualize important actors and periods in the development of each corporation throughout its existence. Despite the elucidative importance, this information was still absent in 12% of the sampled websites (Figure 9). A full explanation was expected, as this information defines all corporations' expectations for the market, as well as being able to generate a timeline of activities. Nonetheless, the spaces delimited for the dissemination of news were present in 60% of the websites (Figure 9). This information may provide the most current and relevant facts on the sector and companies, including the respective main objectives, corporate strategies and specific achievements.

Detailed information on environmental certifications and forest management were identified in 57% and 56% of the websites, respectively (Figure 9). Despite this positive result, several companies still behave like rookies, because they miss the opportunity to clarify and publicize their environmentally friendly practices and results for the market.

Contrary to practices that usually restrict the manufacturing details of products by corporations, there was an evident sharing of production schemes on the manufacture of cross-laminated timber. This representativeness reached 76% of this industry (Figure 9). Other unexpected conditions were identified by the limited approaches to technological innovation and work teams, respectively present in 42% and 36%. Greater detail would reduce these gaps, clarify the technologies in use, and provide greater transparency with respect to the professional structure of these corporations.

The elucidation of cross-laminated timber in its various characteristics as a glued structural product and the construction application stamped a more detailed scenario, which was confirmed by the visible presence of all the topics observed in the fifth topic (Figure 10). All 97 producers shared photographs and images to present and publicize products, whose condition was the only item available on 100% of websites. Positively, CLT products were visibly illustrated, although this divulgation could be intensified.

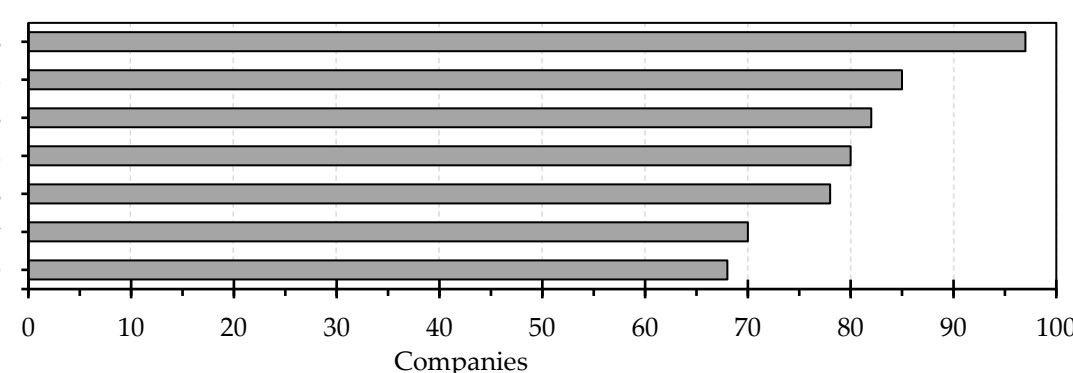

**Figure 10.** Fifth question: details of products shared on the websites of CLT developers. (n = 97 t = March 2023).

Mentions and illustrations of compact and large works were present in 88% and 80% of this population (E = ±0.51%), which indicates that corporations have been attentive to the need to exemplify the applications of their products. In 85% of the companies, the advantages of their products were specified, and 82% of the producers described their products and services in detail (Figure 10). Despite their expressive presence in about 72% of the industry, sustainability and the performance of CLT products were attributes with lower results. On the other hand, 30% of manufacturers still did not take advantage of the opportunities provided by their websites to emphasize these positive characteristics of CLT products, especially at a moment in which corporations have sought competitive advantage for their product lines. This industry should rethink its virtual strategies by using powerful actions to describe products and elucidate environmentally positive features, as addressed by [1].

In the complete panorama identified by these five questions (Figures 6–10), it can be stated that cross-laminated timber is manufactured by an individually representative and territorially expressive industry, which includes 98 manufacturers with 113 factories dispersed around the world (Appendix A) in March 2023, although there is a greater predominance in the northern hemisphere and a perceptible promising presence in the southern hemisphere (Figure 5). In addition, cross-laminated timber is present in those main contents explored by the websites of its producers (Figure 10), being less evident in their corporate structures and goals (Figure 9), and more representative in their forms of communication, either by 24 different languages and multilingual pages (Figure 8a,b) or by numerous types of customer contact channels (Figure 7), including seven different social

networks. Thereby, these sectoral outcomes may suggest the strong ability of this global industry to serve and conquer different international markets.

## 4. Discussion

### 4.1. Current Scenario of the Global Industry of Cross-Laminated Timber

Muszynski et al. [26] previously predicted a global industry potentially formed by 37 companies in 2016, of which only 19 manufacturers were specified by name: KLH, Binderholz, Stora Enso, Hasslacher Norica, Mayr-Melnhof Holz, Pius Schuler, Schilliger Holz, Egoin, X-Lam Dolomiti, Artuso Legnami, Martinsons, StructurLam, Nordic, Merk, DR Johnson, SmartLam, Stephan Holz, Mayr & Sonntag, Eugen Decker, and Novatop. In this presentation, the locations of these companies were specified, including six factories in Austria, four in Germany, two in Switzerland, two in Italy, two in the United States, two in Canada, one company in Spain, and another in Czechia. It was possible to confirm that these CLT developers are still active in March 2023 (Appendix A, Tables A1 and A2). It should be noted that Mayr & Sonntag, although active, was not considered because this company is essentially oriented to the manufacture of cross-nailed laminated timber, which is completely outside the requirements of this research on glued products. During the initial stage, other producers of this type of nailed panel were disregarded as well.

Posteriorly, this global industry could be formed by 66 CLT developers, as forecasted by Larasatie et al. [23] and Muszynski et al. [18,25], although no complete lists have been formally identified in the literature searches. Therefore, the initial stage of this study resulted in unprecedented outcomes, which are specified in Appendix A for 2023.

As only 45 producers formally declared the locations of their factories through their official websites, this study was able to extrapolate the partial results shown in Figure 6 by consulting the different documents consulted in the initial search stage. Therefore, this general overview considered the identifications collected in line with the information shared by [19,27–34]. This consideration allowed the identification of all factories and their locations, which are presented in five tables in Appendix A.

The total amount in Table 5 of 98 manufacturers, specified in Appendix A, showed a business volume perceptibly greater than previous amounts. This sum was exceeded in 62% of the global population and 81% of the specified producers revealed by [26], and in 33% of the volume suggested (although not specified by name) by [18,23].

Fourteen CLT developers have two or more manufacturing facilities (Appendix A). Five North American companies have two plants each, a condition which is already seen in eight European companies. In Europe, a developer has four CLT factories. In addition, four European producers are already categorized as multinational companies, as they have manufactured CLT products in factories located in different countries.

Therefore, the present study was innovative in its formal presentation of a complete list, which specified the name of each manufacturer of cross-laminated timber and the locations of factories which were active in March 2023. This unprecedented list serves to complete the limited sectoral information and evince the global presence of this industry.

The greater concentration of manufacturers in territories above the equator may be justified by economic and social issues. Although there are poor nations in the upper part and some rich countries in the southern subdivision of the planet, Davis [36] verified that the northern hemisphere is marked by many national economies with higher levels of industrialization and income, a condition that is different in the southern hemisphere, where there is a greater predominance of less-developed nations.

Cross-laminated timber was initially created in Europe, and spread across the world from that continent [1,2,8]. This fact justifies a more intense development of the CLT industry in the European region (Table A2), as this continent has been associated with CLT since the period when this engineered wood product was developed in its domains. In addition, the German–Austrian origins of CLT products as stated by [4,7,8] may explain the higher number of facilities in both nations (Appendix A). This scenario still confirms the more favorable environment in which Central Europe (Austria, Germany, Poland, Italy,

Switzerland, and Czechia) became the region with the largest CLT production and market in the world, as stated by [37].

The perceptible presence of CLT developers in the United States (Appendix A1) is related to the greater interest in mass timber products and new manufacturing facilities to satisfy the construction sector, as stated by [38]. In turn, Japan has been actively promoting plantation timber as a sustainable alternative for producing products, which is why a high-value-added timber chain is noted among its eight CLT manufacturers [39]. Furthermore, the proliferation of recent technical codes developed to regulate CLT-based buildings in North America, Europe, Japan, Brazil and South Africa as raised by [1], leads to this more favorable scenario, which may explain some of these larger industrial concentrations and, simultaneously, promote new manufacturing opportunities.

The most evident manufacturing presence in those territories with greater economic and industrial competencies is confirmed in the representation in Figure 5. Of those 26 countries that include at least one local producer, only Ukraine is classified, according to the report on national wealth by the Credit Suisse Research Institute [40], as a lower middle-income nation. The Ukrainian presence would be explained by the long-standing tradition of using wood in construction throughout the eastern European region, as reported by Subtelny [41].

According to the Swiss report [40], Brazil, Argentina and Russia were classified by incomes at the upper-middle level, while 22 other territories specified by Figure 5 are already considered as high-income countries. While Russia is also marked by the same tradition verified in Ukraine by [41], the presence of producers in that locality and also in Brazil and Argentina is justified by the large amount of forest areas in these regions, as reported by The World Bank [42]. The existence of a production sector in different locations oriented to producing different types of timber houses, as identified by [20,43,44], establishes an interesting environment that may contribute to the development of the timber industry, including its engineered products.

In relation to these 22 richer territories, the presence of a timber-construction culture as detailed by [45], the forest availability indicated by [42] and the high level of economic development verified by [40] represent some of the main factors that favor the greater concentration of producers focused on engineered wood products in the more-developed regions, a condition which agrees with the representation in Figure 5. Yet, China has no CLT-focused factories (Figure 5), despite its extensive forests, cited by [42], and its smaller market shown on a few websites using the Chinese language (Figure 8a). Even with its recent code on the design of timber structures [46] and its bamboo cultural heritage materialized by ancient buildings [47], CLT is still incipient in China, compared to lumber, steel and cement.

Regarding the five questions studied in this sectoral survey, it is very important to emphasize that the global CLT industry is becoming a leading asset for the timber-forest chain, either due to the representative participation with an ideal margin of error (E = ±0.51%) or the efficient levels of maturity evinced from the present analysis on the official websites. It is possible to verify that there is still space to improve these platforms, especially through a greater amount of detailed information about the corporations.

The lack of information reported in Figure 6a, which covers 5% of these 97 evaluated producers, is not consistent with the objectives of a competitive corporation, because it was expected that complete addresses would be available. It was observed that 8% of the industry still needs to specify the city and address of their headquarters and, above all, their nations, insofar as this national information was forgotten by 48% of the sample. In addition, the form of presentation of this information represented an identified failure, as locations were found dispersed among different items/tabs of the evaluated websites, which implies a difficulty of understanding the origins and locations of many producers. In response to these flaws, it is suggested that any geographical information be clearly demonstrated on the main page (home page), possibly at the bottom, as well as in the "contacts" item.

Both in terms of quantity and multiplicity, the sharing of corporate profiles on the social networks identified in Figure 7 showed an active connection of this global industry with the general public, which is increasingly present in the virtual channels. The presentation of products through different media (videos, images, and photos) is possible through social networks such as Instagram®, Facebook®, and YouTube®, which is the reason why these tools are popular in this industry. VK® and Xing® were present on a smaller scale, as they are more regionalized solutions, specifically for Russian and Chinese people. The presence of Twitter® was more evident in North American countries, despite the use of this tool in other regions. These platforms may become efficient alternatives for promoting services and products. This condition is also applicable to the use of LinkedIn®, providing a solution with a relevant potential for exploitation by more than half of this industry, as LinkedIn profiles provide visibility for contacting professionals and attracting potential partners and investors.

The current panorama of this studied industry may demonstrate its willingness to attract audiences from different origins, since a wide range of languages is confirmed in Figure 8a. About a third of the websites already offered three or more languages, with 52% already sharing two or more languages (Figure 8b). Apart from the people who are fluent in two or more distinct languages (Canada, Switzerland, Belgium, South Africa, and Ukraine) and countries that were receptive to immigrants fluent in other languages (the United States, Brazil, Germany, Austria, Australia, the United Kingdom, France, Spain, Turkey, Sweden, and Norway), 11% of the official websites shared five or more languages. This plural portion may be related to a greater openness and willingness of companies to attend, in addition to internal markets, to foreign markets.

It was expected that the presentation of actions and businesses, goals and visions, and historic timelines would be contents disseminated by all companies. However, the reality showed that some companies still fail to elucidate these topics, a situation which leads to a lack of clarity and objectives regarding the corporate strategies. The production schemes and practices represented an abnormality, as this sharing at substantial levels was not expected (Figure 9).

For reasons of industrial confidentiality and market reserve, it is common practice for industrial corporations to restrict information about the ways and tools used in the manufacture of their products. This situation was contrary to what was observed. Although more than half of the companies shared information about their environmental certifications and forest management, a large portion still did not utilize the benefits of these clarifications on their official websites. Given the environmental advantages of cross-laminated timber in construction, as identified and addressed by [1], this condition contributed to hiding several benefits related to this industry, which is evidently greener and more sustainable than other industrial sectors driven by the consumption of mineral resources and with negative environmental impacts.

In terms of product detailing (Figure 10), all options were surpassed in 70% (or more) of the population. The wide dissemination of product attributes was positive, especially for the attraction of new customers and partners. This detailed overview of the vocations and attributes of cross-laminated timber can contribute to a more assertive understanding of its advantages, forms, uses, specifications, and sustainable and strength features. This industry may be inspired by [1], using information from that review of CLT products.

*4.2. Future Perspectives of the Global Industry of Cross-Laminated Timber*

Although the global CLT industry has a growth perspective that could exceed the four-million level by 2025, its annual production recorded in 2021 is just below three million cubic meters [12]. Of this amount, the European industry contributed just over 40%, exceeding one million cubic meters of production [33]. The CLT participation in the product lines of some of the main wood industries in Europe was detailed by Table 6.

**Table 6.** Average annual production reported by some of the main wood industries in Europe.

| Company | Country | Lumber Production [1] (Cubic Meters) | CLT Production [1] (Cubic Meters) | CLT Participation [1] (%) |
|---|---|---|---|---|
| Binderholz | Austria | 4,430,000 | 220,000 | 5% |
| Stora Enso | Finland | 5,700,000 | 180,000 | 3% |
| KLH | Austria | – | 150,000 | – |
| Mayr Melnhof Holz | Austria | 1,700,000 | 75,000 | 4% |
| Hasslacher/Nordlam | Austria | 600,000 | 70,000 | 14% |
| Pfeifer Holz | Germany | 2,200,000 | 60,000 | 3% |
| HBS Berga/Ante-holz | Germany | 1,490,000 | 55,000 | 4% |
| Theurl Austrian Premium | Austria | – | 45,000 | – |
| Schilliger Holz | Switzerland | 600,000 | 40,000 | 7% |
| Derix Gruppe | Germany | – | 40,000 | – |
| Züblin Timber | Germany | – | 30,000 | – |
| Artuso Legnami | Italy | – | 30,000 | – |
| Lignotrend | Germany | – | 28,000 | – |
| Eugen Decker | Germany | – | 25,000 | – |
| Ziegler Holzindustrie | Germany | 1,210,000 | 20,000 | 2% |

[1] values calculated by the authors through the production data available in [19,32,48,49].

Comparing the recent data from the European region, it was found that only three companies had consecutive CLT production of above 100,000 cubic meters per year, with four companies exceeding 50,000 m$^3$, and three other companies exceeding 40,000 m$^3$. The representativeness of CLT products in wood processing activities is still proportionally low, compared to the amount of wood processed (Table 6). The scenario suggests an interesting potential for increasing CLT production and market, since only one company surpassed the concentration of 10% of its operations destined for this purpose.

This fact agrees with the recent observations of [1], which state that there is a great market opportunity for the CLT production on a global scale, especially in supplying distinct demands through prefabricated and modular buildings. In practice, Hosseini and Peer [50] verified that wood processing in sawmills should be established, using the opportunities to increase sawmill profitability and modernize plants using automation, energy efficiency, environmentally friendly production, and a line of variable products. On the other hand, relations among Industry 4.0, construction modernization and CLT production still need to be analyzed by new studies and intensified by global sectors [51]. Overall, new plants are required in regions with few or no developers, as these scenarios lead to market dominance and higher prices caused by less competition, as cited by [45].

Another factor that is attracting greater interest is verified by the predicted increase in current CLT production in the near future, at least in the countries of the northern hemisphere. Jauk [19] predicts that European production will reach two million cubic meters in 2023. The increase will be easily achieved when new plants come into operation in 2024 (Table 7). The inclusion of CLT in emerging markets, regardless of its intrinsic conditions in relation to their macroeconomic scenarios, would demand the use of new policies and economic incentives to boost the sylviculture- and timber-based sectors. As suggested by [1], new policies could consider CLT products in the expansion of housing.

**Table 7.** Production capacities and specificities of future manufacturing plants.

| Company | Country | Additional Capacity [1] (Cubic Meters) | Implementation [1] (Period) | Factory Specificities [1] |
|---|---|---|---|---|
| Ziegler Holzindustrie | Germany | 150,000 | 2023 | 2 new lines |
| Mayr-Melnhof Holz | Austria | 140,000 | 2023 | 1 expansion + 1 new plant |
| KLH | Austria | 140,000 | 2023 | 2 expansions in 2 plants |
| Stora Enso | Sweden | 120,000 | 2023 | 1 new plant |
| Mosser Leimholz | Austria | 55,000 | 2023 | 1 expansion (plant/sawmill) |
| Schilliger | Austria | 100,000 | 2023 | 1 renovation + 1 new plant |
| Holzbauwerk Schwarzwald | Germany | 35,000 | 2023 | 1 new plant |
| Hasslacher/Nordlam | Austria | 50,000 | 2023 | 1 new combined line |
| Smartlam North America | United States | 185,000 | 2024 | 1 plant expansion |
| Best Wood Schneider | Germany | 100,000 | 2024 | 1 expansion + 1 sawmill |
| LOC Holz | Austria | 45,000 | 2024 | 1 new plant |
| Holzwerke van Roje | Germany | 75,000 | 2024 | 1 expansion |
| LignaTerra | United States | 28,000 | 2024 | 1 new plant |
| CLTech | Germany | 25,000 | 2024 | 1 new plant |

[1] data available in [31,33].

Moreover, the main attention may be on prioritizing scientific research to identify the most appropriate silvicultural species for each territory and to study the native species most suitable for conversion into engineered products. Even so, it is possible to consider some nations with the best future potential for the development of the production and market of engineered wood products and, consequently, the installation of new factories and efficient manufacturing lines to produce modern inputs such as CLTs.

From the economic data reported by [40] in line with the presence of the massive forests stated by [42] and with the traditionally and recently industrialized nations cited by [52], it should be noted that some regions present more significant potential, given those axes formed by the availability of bioresources and economic–industrial resources, to install future factories aimed at producing CLT products. This group would include the Netherlands, Denmark, Croatia, Portugal, Lithuania, Slovakia, Slovenia, Hungary, Bulgaria, Turkey, Mexico, China, Thailand, Malaysia, Indonesia, South Korea, India, and New Zealand. Colombia, Uruguay, Paraguay, Peru, Ecuador, Tunisia, Gabon, Botswana, and Ghana may be added if their domestic powers and conditions are improved. In line with this, awareness of timber buildings is required to guide local professionals [53,54].

## 5. Conclusions

The systematic review and different literature searches identified few studies about the cross-laminated timber industry, which rarely detailed the names and locations of manufacturers. A list was formally developed, which symbolized a practical finding for this global industry. This outcome shared unprecedented information with the literature, being essential for the wood chain, including the forest, lumber and construction sectors.

The updated list identified the name of each active producer, quantified the effective number of manufacturers comprising 98 players in operation in March 2023, and detailed the 113 active manufacturing plants spread over 26 different countries spread over the five habitable continents of our planet: Europe, America, Oceania, Africa, and Asia. It should be noted that this list must present changes over time, in the face of corporate dynamism, as developers are created, closed, merged with each other or even expanded into manufacturing facilities previously dedicated to other engineered wood products. The importance of specifying and designating all developers becomes evident as a formal representation of those manufacturing activities operating in 2023. The final stage of this

virtual-based study rigorously recorded a global panorama of the glued cross-laminated timber industry, identified by a representative sectoral survey using corporate websites.

The sectoral survey evidenced a perceptible industry with regard to the number of developers and territories with manufacturing plants, as its geographical representation included a global production coverage in both hemispheres. This representativeness was maintained in the sampling process, which considered 97 of the 98 producers, since the absent manufacturer was not evaluated because its official website was not found.

Even so, the analysis of official websites showed that this industry was already exposed to different populations, either by the global distribution of more than a hundred factories or by the total of 24 languages available on the websites analyzed. In addition, official profiles in different social networks (Facebook®, Instagram®, LinkedIn®, YouTube®, Twitter®, Xing® and VK®) were also identified in this sectoral survey, through official websites, which revealed a multiplatform virtual presence. Glued CLT products represented the most detailed content of the sampled producers, including textual and graphic information, which suggests an effective predisposition of this industry to conquer and supply both regional and foreign markets, through diversified languages and channels for customers.

From these outcomes, the development of the cross-laminated timber industry may be confirmed by the present existence of four multinational companies, as well as by the global expansion of production capacities through some implementations of new plants and modernizations of manufacturing lines. These positive facts, combined with the multiple advantages and applications of CLT products, reinforce the global motivation and interest of the construction industrialization in using more rational and sustainable manufactured solutions, strongly based on greener engineering products. Further studies may continue this observation in order to keep the content up to date, develop policy to boost markets through the greater consumption of CLT products, develop new sectoral surveys to identify domestic particularities, design new plans and case studies to install manufacturing facilities in nations with timber-forest potential and adequate economic perspectives, and analyze the scenario of completed and ongoing CLT-based projects.

**Author Contributions:** All stages, V.D.A. and A.C. All authors have read and agreed to the published version of the manuscript, which is part of V.D.A.'s postdoctoral study completed at UFSCar in 2023. All authors have read and agreed to the published version of the manuscript.

**Funding:** This research received no external funding. It was completed in 1 year of work, using the authors' own resources and efforts.

**Institutional Review Board Statement:** Not applicable.

**Informed Consent Statement:** Not applicable.

**Data Availability Statement:** Data available on request, due to privacy restrictions.

**Conflicts of Interest:** The authors declare no conflict of interest.

## Appendix A

For sectoral listing, the results were organized alphabetically by company names, according to the continent of the CLT factory location in March 2023 (Tables A1–A5).

**Table A1.** Developers of cross-laminated timber and locations in the American continent.

| Company Name | City (Country) of Each Manufacturing Plant |
| --- | --- |
| Arauco | Arauco (Chile) |
| Bell Structural Solutions/Alamco | New Brighton and Albert Lea (United States) |
| Crosslam | Suzano (Brazil) |
| DR Johnson | Riddle (United States) |
| Element5 | Ripon and Saint Thomas (Canada) |

**Table A1.** *Cont.*

| Company Name | City (Country) of Each Manufacturing Plant |
| --- | --- |
| Euclid Timber Frames | Heber City (United States) |
| Kalesnikoff | South Slocan (Canada) |
| Mercer Mass Timber | Spokane (United States) |
| Nordic Structures | Chibougamau (Canada) |
| Novak CLT | Neuquén (Argentina) |
| Smartlam North America | Columbia Falls and Dothan (United States) |
| Sterling | Phoenix and Lufkin (United States) |
| StructureCraft | Abbotsford (Canada) |
| Structurlam Mass Timber | Penticton (Canada) |
| Texas CLT | Magnolia (United States) |
| Urbem | Almirante Tamandaré (Brazil) |
| Vaagen Timbers | Colville (United States) |
| Western Archrib | Edmonton and Boissevain (Canada) |

**Table A2.** Developers of cross-laminated timber and locations in the European continent.

| Company Name | City (Country) of Each Manufacturing Plant |
| --- | --- |
| Agrop Nova/Novatop | Pteni (Czechia) |
| Ante-holz/HBS Berga | Bromskirchen and Berga (Germany) |
| Arcwood | Polva (Estonia) |
| Artuso Legnami | Caselle di Altivole (Italy) |
| Belliard | Gorron (France) |
| Best Timber Polska | Torun (Poland) |
| Best Wood Schneider | Messkirch (Germany) |
| Binderholz | Burgbernheim (Germany) and Fugen (Austria) |
| Buckland Timber | Crediton (England) |
| Cedarlam | Cork (Ireland) |
| CLT Profi | Jelgava (Latvia) |
| CLT Prom | Moscow (Russia) |
| CLT Rus | Moscow (Russia) |
| CLT Suisse | Orges (Switzerland) |
| CrossLam Kuhmo | Kuhmo (Finland) |
| Damiani Legnami | Brixen (Italy) |
| Derix Gruppe | Niederkrüchten and Westerkappeln (Germany) |
| Egoin | Bizkaia (Spain) |
| Essepi | Cavedine (Italy) |
| Eugen Decker | Morbach (Germany) |
| FHS Holzbau | Grassau (Germany) |
| Gauye & Dayer | Sion (Switzerland) |

**Table A2.** *Cont.*

| Company Name | City (Country) of Each Manufacturing Plant |
|---|---|
| Glenfort Timber Engineering | Dungannon (Ireland) |
| Grossmann | Rosenheim (Germany) |
| Hasslacher/Nordlam | Stall (Austria) and Magdeburg (Germany) |
| Hoisko/CLT Finland | Hoisko (Finland) |
| Holzbau Unterrainer | Ainet (Austria) |
| HolzBauWerk Schwarzwald | Seewald-Besenfeld (Germany) |
| Holzwerke van Roje | Oberhonnefeld-Gierend (Germany) |
| Jekabpils PMK | Jekabpils (Latvia) |
| KLH | Teufenbach-Katsch and Wolfsberg (Austria) |
| Kurt Huber | Achern (Germany) |
| Lignotrend | Weilheim-Bannholz (Germany) |
| LOC Holz | Arbing (Austria) |
| Martinsons/Holmen | Bygdsiljum (Sweden) |
| Mayr Melnhof Holz | Gaishorn and Reuthe (Austria) |
| Merkle Holz | Nersingen-Oberfahlheim (Germany) |
| Moser Holzbau | Taisten (Italy) |
| Mosser Leimholz | Randegg (Austria) |
| Nema | Olesnice (Czechia) |
| Nemétona | Cuenca (Spain) |
| Nordic CLT | Aluksne (Latvia) |
| Ortner Holz | Tragwein (Austria) |
| Pfeifer Holz | Schlitz (Germany) |
| Pius Schuler | Rothenthurm (Switzerland) |
| Piveteau Bois | Sainte Florence (France) |
| Rezult | Korosten (Ukraine) |
| Rubner Holzbau | Brixen (Italy) and Ober-Grafendorf (Austria) |
| Sägewerk Meißnitzer | Niedernsill (Austria) |
| SBS | Friesenheim (Germany) |
| Schilliger Holz | Haltikon (Switzerland) |
| Schmid Holzbau | Bobingen (Germany) |
| Sebastia | Lleida (Spain) |
| Segezha Group/Sokol | Vologda (Russia) |
| Setra | Langshyttan (Sweden) |
| Skonto/Cross Timber | Jelgava (Latvia) |
| Södra | Varberg (Sweden) |
| Splitkon | Amot (Norway) |
| Stabilame | Mariembourg (Belgium) |
| Stora Enso | Gruvon (Finland), Zdirec (Czechia), Ybbs and Bad St. Leonhard (Austria) |
| Tanguy/Tot'm | Brest (France) |
| Theurl Austrian Premium | Steinfeld (Austria) |

**Table A2.** *Cont.*

| Company Name | City (Country) of Each Manufacturing Plant |
| --- | --- |
| Weinberger | Reichenfels (Austria) |
| Wigo by Praslas | Limbazi (Latvia) |
| Xilonor | La Coruňa (Spain) |
| XLam Dolomiti | Castelnuovo (Italy) |
| Ziegler Holzindustrie | Hermsdorf (Germany) |
| Zimmerei Wolf | Reichenbach (Germany) |
| Züblin/Merk Timber | Aichach and Gaildorf (Germany) |

**Table A3.** Developers of cross-laminated timber and locations in the Asian continent.

| Company Name | City (Country) of Each Manufacturing Plant |
| --- | --- |
| Chuto | Ishikawa (Japan) |
| Cypress Sunadaya | Ehime (Japan) |
| JKHD Holding/Okhotsk Wood Pia | Hokkaido (Japan) |
| Meiken Kogyo | Okayama (Japan) |
| Seihoku Plywood | Miyagi (Japan) |
| Timberam | Akita (Japan) |
| Tottori CLT | Tottori (Japan) |
| Yamasa Mokuzai | Kagoshima (Japan) |

**Table A4.** Developers of cross-laminated timber and locations in the African continent.

| Company Name | City (Country) of Each Manufacturing Plant |
| --- | --- |
| Xlam South Africa | Cape Town (South Africa) |

**Table A5.** Developers of cross-laminated timber and locations in the Oceanian continent.

| Company Name | City (Country) of Each Manufacturing Plant |
| --- | --- |
| Mayflower Group/Xlam | Wodonga (Australia) |

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
