# Peer review of "The Global Cross-Laminated Timber (CLT) Industry: A Systematic Review and a Sectoral Survey of Its Main Developers"

_sustainability, doi:10.3390/su15107827_

Round 1

Reviewer 1 Report

CLT is a major development for construction sector and a fashionable object for architects, structural designers and increasing interest for scientist worldwide. The authors have performed an interesting contribution in MDPI Forest 2023 reminded in reference #1 of this contribution (DOI:10.3390/f14020264) leading much more as a benchmarking than a scientific advancement.

Reference #2 is indicating a publication on fire resistance of CLT and not global function of the product as indicated. It should be pointed out here Brandner_2016 review for example. Reference #9 is not specifically focused on CLT LCA, this should be modified. Generally, references are not pointing out major contributions of the sector.

Reviewer is not convinced as global semantics search in online web-database can be considered as a Material & Method tool but for inter-disciplinary journal as proposed, this question can be discussed.

Table 1 is showing general strings that have been searched. It’s the same keywords as proposed in MDPI Forest DOI:10.3390/f14020264 but leading to a completely different result presented here in figure 1a (about some tenth per year) compared to about some thousands per year previously published (figure 1a). The difference between those 2 approaches is not at all enough clear to the reader.

Reviewer don’t see any interest to point out 1st author of a publication index presented in figure 2 ; moreover missing some major colleagues in the field (in which I am not considering to be a part of). This review is insufficient and should have been largely extended.

Survey presented worldwide in figure 5 is interesting but should have been correlated to country surface, country volume of wood produced by forest harvesting. It’s just a bit aborded in table 9 but without discussion.

Reviewer don’t see an interest to customer channels available, number of languages, social networks, etc without major discussion on this information for the sector. The questioning of vision and goals (figure 9) requested is not detailed on their criterions to evaluate it.

The interesting point is in 1st sentence of the discussion and annexes table A2 is a list of CLT producers, but that’s it.

Reviewer recognize the effort made to write this detailed contribution but do not considered it as a valuable publication as it is proposed.

Author Response

CLT is a major development for construction sector and a fashionable object for architects, structural designers and increasing interest for scientist worldwide. The authors have performed an interesting contribution in MDPI Forest 2023 reminded in reference #1 of this contribution (DOI:10.3390/f14020264) leading much more as a benchmarking than a scientific advancement.

R: We really appreciate your efforts in this reviewing process. Your perception summarizes the continuity of our observations, now, about the industry.

Reference #2 is indicating a publication on fire resistance of CLT and not global function of the product as indicated. It should be pointed out here Brandner_2016 review for example. Reference #9 is not specifically focused on CLT LCA, this should be modified. Generally, references are not pointing out major contributions of the sector.

R: Thank you. Reference #2 was corrected (now #3), as we emphasized this fire behavior. About #9, the sentence is not about LCA. Even so, we rewrote it to emphasize the findings identified by #9. Of course the references are not about the sector, which justified the novelty of the present approach due to the lack of sectorial data – which was identified and discussed by our manuscript.

Reviewer is not convinced as global semantics search in online web-database can be considered as a Material & Method tool but for inter-disciplinary journal as proposed, this question can be discussed.

R: Authors are not convinced as this specific opinion without any justification or substantiation could be taken seriously. At the current moment marked by a global pandemic and the intensification of internet searches, alternative virtual-based methodologies and modern research protocols are being established to obtain scientific studies with very efficient levels of repetition. A good example of this statement is confirmed in the present study, specifically on the high level of reliability justified by the representative sampling obtained with a broader observation of the studied industry on global level.

Table 1 is showing general strings that have been searched. It’s the same keywords as proposed in MDPI Forest DOI:10.3390/f14020264 but leading to a completely different result presented here in figure 1a (about some tenth per year) compared to about some thousands per year previously published (figure 1a). The difference between those 2 approaches is not at all enough clear to the reader.

R: We believe that the reviewer did not pay due attention to the strings of both studies, since they are completely different. For an obvious reason of the main research scopes on CLT, only main term "cross-laminated timber" was repeated in both studies. However, they differed from each other in the further keywords cited in both strings, insofar as the first study considered terms related to construction topic and the present study was dedicated to industry approach. Due to different sides, of course results will be different. Any reader will identify by the title that there are different approaches of both studies, as the first paper addressed CLT product and its function through a systematic review and, therefore, the present article is a continuation of this initial study with a specific approach to evaluate CLT industry under global levels, including another specific systematic review and, obviously, sectorial survey about the aforementioned industry.

 Reviewer don’t see any interest to point out 1st author of a publication index presented in figure 2 ; moreover missing some major colleagues in the field (in which I am not considering to be a part of). This review is insufficient and should have been largely extended.

R: Once again, our contribution is specifically dedicated to study CLT industry and respective developers, which is why only rare contributions within this scope were duly regarded. Despite the importance of other contributions on CLT (e.g., about product development, alternative processes, new manufacturing, material testing, structural design, etc.), such topics are not relevant for sectorial surveys, insofar as this study is specifically dedicated to understand the current existence, coverage, goals and scope the studied industry. Therefore, the importance of this study is unique, since there are no global studies, especially for an industrial sector that is so recent and little studied. Thus, other studies related to distant topics are not regarded, which is why many colleagues are not specified in that figure, which is only focused on those colleagues with studies specifically on CLT industry identified in multiple prospections. In addition, our systematic review proved the novelty of approach, as few studies were identified using different strings in two different databases (Scopus and Web of Science).

Survey presented worldwide in figure 5 is interesting but should have been correlated to country surface, country volume of wood produced by forest harvesting. It’s just a bit aborded in table 9 but without discussion.

R: Reviewer really did not realize that we authored this figure 5, which makes it impractical/inconvenient to insert data from other sources as it is one of our findings. Although we did not specify volumes, we led our discussions based on countries with the largest forest areas as you suggest. Discussion approached economic and forest arguments to justify the presence of this industry in those regions. You may check it through “The greater concentration of producers in territories above the Equator line may be justified by economic and social issues. Although there are poor nations in the upper part and some rich countries in the southern subdivision of the planet, Davis [37] verified that the northern hemisphere is marked by many national economies with higher levels of industrialization and income, a condition that is different in the southern hemisphere, where there is a greater predominance of less developed nations. Cross-laminated timber was initially created in Europe, and spread across the world from that continent [1,2,8]. This fact justifies a more intense development of CLT industry in European region (Table A2), as this continent has been associated with CLT since the period when this engineered wood product was developed in its domains. In addition, the German-Austrian origins of CLT products as stated by [4,7,8] may explain the higher number of facilities in both nations (Appendix A). This scenario still confirms the more favorable environment in which the Central Europe (Austria, Germany, Italy, Czechia, Switzerland, and Poland) became the region with the largest CLT production and market in the world as stated by [38]. The perceptible presence of CLT developers in the United States (Appendix A1) is related to the greater interest in mass timber products and new facilities to satisfy construction sector as stated by [39]. In turn, Japan has been actively promoting plantation timber as a sustainable alternative to produce products, which is why a high-value added timber chain is noted through its 8 CLT manufacturers [40]. Furthermore, the proliferation of recent technical codes developed to regulate CLT-based buildings in the North America, Europe, Japan, Brazil and South Africa as raised by [1], leads to this more favorable scenario that may explain some of these larger industrial concentrations and, simultaneously, promote new manufacturing opportunities. The most evident manufacturing presence in those territories with greater economic and industrial competences was confirmed in the representation of Figure 5. Of those 26 countries that include at least a local manufacturer, only Ukraine is classified, according to the report on national wealth by the Credit Suisse Research Institute [41], as a nation with low-middle economic income. The Ukrainian presence would be explained by the long-standing tradition of using wood in construction throughout the eastern European region, as reported by Subtelny [42]. According to the Swiss report [41], other three nations, Brazil, Argentina and Russia, were classified by incomes at the upper-middle level, while other 22 territories specified by Figure 5 are already considered as high-income countries. While the Russia is also marked by the same tradition verified in Ukraine by [42], the presence of producers in that locality and also in Brazil and Argentina is justified by the large amount of forest areas in these regions, as reported by The World Bank [43]. The existence of a production sector in different locations oriented to producing different types of timber houses, as identified by De Araujo et al. [21,44,45], establishes an interesting environment that may contribute to the development of the timber industry, including its engineered products. In relation to these other 22 richer territories, the presence of a timber culture in construction as detailed by [46], the forest availability indicated by [43] and the high level of economic development verified by [41] represent some of the main factors that favor the greater concentration of producers focused on engineered wood products in the more developed regions, a condition which agrees with the demonstration evinced by Figure 5.”

Reviewer don’t see an interest to customer channels available, number of languages, social networks, etc without major discussion on this information for the sector. The questioning of vision and goals (figure 9) requested is not detailed on their criterions to evaluate it.

R: Although this opinion is something specific to the reviewer, it should be noted that self-knowledge of any industrial sector starts from the analysis of its business information, whether basic or specific, to understand the present scenario and, as presented in the manuscript, indicate different perspectives and conditions of the individuals. Transnational coverage was verified through the utilization of multilingual websites. The evaluation of customer channels and social networks identified a plural presence in many types of communication platforms, which is important in the current moment of the global connectivity. Vision and goals is not a questioning, as they are an evaluation alternative of fourth question, whose presence/absence was identified to characterize the breadth and detail of every corporate history.

The interesting point is in 1st sentence of the discussion and annexes table A2 is a list of CLT producers, but that’s it.

R: Any scientific work is not carried out to please a specific opinion/people, as it must regard and satisfy a large number of individuals. Thereby, the reviewer's contrary opinion is limited to his own thinking that does not consider the eager scenario for sectorial studies, as raised in the present study. Anyway, we thank the reviewer about the discussion contents and tables in the annex (which are unprecedented).

 Reviewer recognize the effort made to write this detailed contribution but do not considered it as a valuable publication as it is proposed.

R: We understand this specific opinion, which differs from the other reviewers, and we emphasize that the reviewer may seek to read more about the practical importance of sectorial studies for companies, which have effective application for industries in the same way as the studies of product/process, since a sectorial survey address different perspectives as led by our contribution, detailing scenarios, unknown conditions, geographic coverages, challenges, future panoramas, market expectations/possibilities, and main commercial products and solutions. Despite negative opinions, we appreciate your suggestions and the confirmation of our study presents a “detailed contribution”.

Reviewer 2 Report

Thank You for your contribution. Is important and interest but i could mentioned that you could be additions it. Is it overviwer articel or research artical?

In topic materials and methods is small mentioned another methodology as yours.

I don´t to find a mention of how long data was collected for the article. Was it after the referenc [21] ?

Tables are mish mash. Table 12 listed in the text in chapter 2.2 it is not in the articel showing. this table is table A2?

Author Response

Thank You for your contribution. Is important and interest but I could mentioned that you could be additions it.

R: We really appreciate your efforts in this reviewing process.

Is it overviwer articel or research artical?

R: Although our research has identified an overview scenario, we did not specify a particular issue as it is expected in this type of study. Five distinct contexts were addressed to detail different aspects (product, location, website, language and client contact) of each analyzed company. The study is a research paper, as it has different contributions to address the subject, as the paper started from a systematic review to show the current scenario in the literature, an unprecedented listing formed by CLT developers identified in exhaustive prospections (Appendix), and a sectorial survey with this population to evince the different aforementioned aspects through a very representative sampling process, reason why our study becomes a scientific contribution with relevant outcomes for the global industry.

 In topic materials and methods is small mentioned another methodology as yours.

R: Small? Three pages were written to detail step-by-step, as the research was organized into two main methods (systematic review and sectorial survey) and observations as explained in the 2.1 section. Also, 2.2 and 2.3 sections were developed to address the prospecting process and its steps, whereas 2.4 and 2.5 sections specified on the sectorial survey process and its steps and the structured script utilized in the sectorial analysis after the complete exhaustive prospection to identify companies described in the 2.4 section. Due to the unprecedented approach, many parts and considerations were initially designed and developed by the authors.

I don´t to find a mention of how long data was collected for the article. Was it after the referenc [21] ?

R: Our analyses occurred in early 2023, however, we specified better in each methodological section. You may verify that, due to some suggestions of another reviewer, we needed to update the list and results. Thus, the period of the new updated analysis was only during March 2023, although prospections were carried out in 2022 and 2023. We inserted this information after reference [21] as suggested by you.

Tables are mish mash. Table 12 listed in the text in chapter 2.2 it is not in the articel showing. this table is table A2?

R: Tables are not mish mash!! It was only an isolated typo. We revised all mentions. Thank you.

Reviewer 3 Report

The manuscript presents an overview of the Cross-Laminated Timber (CLT) Global Industry, with a focus on the participating companies and the various industrial applications.

Overall, the manuscript is well-written and explores an interesting topic with some valuable findings. However, there are important concerns that need to be addressed, which are outlined in detail below.

1. The manuscript could benefit from a more systematic introduction to the topic. It currently seems that the Introduction section starts with a literature review without a preliminary discussion that defines the problem, presents a graphical representation which clarifies the reviewed system, and outlines the research foci. The introduction section could be reorganized for better coherence.

2. In Section 1, the authors mention that two issues exist, but they present three bulleted points, which are misleading for readers.

3. The manuscript contains typos and grammatical mistakes that need to be addressed (some examples: EX1. in “The condition that companies does not value” the verb should be changed to “do not” , and EX2: the word “Generalist” should be replaced with “General”).

4. The manuscript places too much emphasis on the number of works published by authors in the field when selecting important references. Rather, a better approach to evaluating the significance of studies would be to consider their quality based on the number of citations they have received.

5. The difference between the general and specific terms presented in Tables 1 and 2 is unclear, as both categories include specific terms related to cross-laminated timber.

6. Wood products management and optimization, including the participation of developers, has been the subject of several studies and surveys. An important aspect of wood production is improving efficiency through optimization and automation, which has been discussed in recent studies such as https://doi.org/10.3390/automation3030023 and https://doi.org/10.1109/ACCESS.2022.3223053 (references that could be cited in the manuscript). It would be valuable to incorporate this aspect into the current study and explore how companies are addressing these renovations towards more profitable businesses.

7. Did putting a hyphen between words (e.g., cross-laminated timber", "cross laminated timber", and “Xlam”, “X-lam”) return you different references?

8. Manuscript devotes excessive attention to the search methodology, rather than prioritizing the examination of how different categories of research in the CTL industry are receiving attention. Providing information for these statistics is more important than merely distributing papers among publishers, etc.

9. It is a bit strange that Figure 1.b does not include any papers published before 2018.

10. If the maximum number of papers for each author in Figure 2 does not exceed 5, there is no need to plot the horizontal axis scale up to 30.

Author Response

The manuscript presents an overview of the Cross-LaminatedTimber (CLT) Global Industry, with a focus on the participatingcompanies and the various industrial applications. Overall, the manuscript is well-written and explores an interesting topic with some valuable findings.

R: We really appreciate your efforts in this reviewing process. Yes, we carried out a very representative sectorial survey to understand the presence and current contexts of this global modern industry.

However, there are important concerns that need to be addressed, which are outlined in detail below.

  1. The manuscript could benefit from a more systematic introduction to the topic. It currently seems that the Introduction section starts with a literature review without a preliminary discussion that defines the problem, presents a graphical representation which clarifies the reviewed system, and outlines the research foci. The introduction section could be reorganized for better coherence.

R: A long introduction should not complete much of what has been said, especially as the present study is a continuation of an in-depth analysis of CLT products recently published by Forests journal. In addition, a systematic review as led in the present paper, which was intensely utilized to complete and compare results and deepen discussions. Considering that the studied topic was identified as a literature gap by our previous paper, this information was clarified in detail in the introduction to complete justifications.

Even so, the introduction was slightly adjusted as you can check.

  1. In Section 1, the authors mention that two issues exist, but they present three bulleted points, which are misleading for readers.

R: You are right. It was our fault. Issues were rewritten and kept. This number was corrected too.

  1. The manuscript contains typos and grammatical mistakes that need to be addressed (some examples: EX1. in “The condition that companies does not value” the verb should be changed to “do not” , and EX2: the word “Generalist” should be replaced with “General”).

R: You are right. It was our fault. We corrected both terms. Thank you.

  1. The manuscript places too much emphasis on the number of works published by authors in the field when selecting important references. Rather, a better approach to evaluating the significance of studies would be to consider their quality based on the number of citations they have received.

R: For your attention, any complete observation must initially prioritize the volume of publications to identify how 'hot' or 'new' the topic is for science. A contrast in the number of publications was evinced between the general context on CLT (product) and the specific context on the CLT industry addressed in the sectorial survey, reason why the existence of few works really justifies the originality of our approach about the CLT industry (our main goal). In this observation, only 4 papers were identified by Scopus and 2 papers by WoS. But, only a conference paper satisfied the expectations of our contribution. As this single contribution does not have any citation, no mention about citations is necessary.

  1. The difference between the general and specific terms presented in Tables 1 and 2 is unclear, as both categories include specific terms related to cross-laminated timber.

R: We really appreciate your suggestion. While the general searches were carried out using individual terms, the specific condition prioritized the specific expression studied in our research (CLT industry), which is why very different scenarios were identified. While the general searches delivered publications without direct relation to our goal, the second level specifically shared us some documents within the main goal.

  1. Wood products management and optimization, including theparticipation of developers, has been the subject of several studies and surveys. An important aspect of wood production is improving efficiency through optimization and automation, whichhas been discussed in recent studies such as

https://doi.org/10.3390/automation3030023 and https://doi.org/10.1109/ACCESS.2022.3223053

(references thatcould be cited in the manuscript). It would be valuable to incorporate this aspect into the current study and explore howcompanies are addressing these renovations towards moreprofitable businesses.

R: Thanks for your tips. These topics are really important for the wood production, however, they are out of place in sectoral analyzes aimed at understanding the geographic location, available products, website communication, and market coverage established by the languages offered in their websites. Of course they are important topics, but they could be a better fit in studies to analyze and develop manufactures, production systems, and process modernization. About both studies, they are important, but they do not address a sectoral analysis of CLT developers. Even so, both studies were cited to complete discussions.

  1. Did putting a hyphen between words (e.g., cross-laminatedtimber", "cross laminated timber", and “Xlam”, “X-lam”) return you different references?

R: As the literature and market utilize both terms (justified by differences in English languages from England and USA), we decided to utilize both as a strategy to obtain an efficient filtering process and avoid the elimination of any publication as we utilized different databases to prospect literature.

  1. Manuscript devotes excessive attention to the search methodology, rather than prioritizing the examination of how different categories of research in the CTL industry are receiving attention. Providing information for these statistics is more important than merely distributing papers among publishers, etc.

R: If our study was only a systematic review, possibly this way should be more effective to present each distinct goal and provide statistics about each field of research. As these results are only from the general view, which was not prioritized, our goal was dedicated to understand the industry through the second stage of our methodology, that is, the sectorial survey. Therefore, this suggestion would not bring strong contributions to this work in question, because the filtering process aimed to identify the rare studies on the global industry and its developers instead of the study areas portrayed in other studies.

  1. It is a bit strange that Figure 1.b does not include any papers published before 2018.

R: Strange? In practice, it is the result of the total absence of publications exclusively dedicated to address the CLT industry, which justifies our proposal and sectorial survey using very representative and unprecedented ways.

  1. If the maximum number of papers for each author in Figure 2 does not exceed 5, there is no need to plot the horizontal axis scale up to 30.

R: You are right. We adjusted the axis scale. Thank you.

Reviewer 4 Report

In general, this review study is of interest to stakeholders in mass timber industry by reporting on the status of the facilities at a global scale. It would add scholarly value if the volume of CLT production is compared in addition to the number of facilities on each continent. Although Table 6 provided some production volume data, it would have been a valuable addition in the result section.

The state of CLT industry development is closely associated with the time period in which CLT started to developed. Europe implemented CLT earlier than other countries, which explains the higher number of facilities there. The authors should address this aspect either in the review methodology or discussion.

Author Response

In general, this review study is of interest to stakeholders in mass timber industry by reporting on the status of the facilities at a global scale. It would add scholarly value if the volume of CLT production is compared in addition to the number of facilities on each continent. Although Table 6 provided some production volume data, it would have been a valuable addition in the result section.

R: We really appreciate your efforts in this reviewing process. Unfortunately, production volumes are not usually shared by companies. The novelty of our approach raised the first listing of CLT producers, which shows how unprecedented is this subject. Due to corporate strategies, few developers have disclosed own product capacity and production volumes, being a very complex data to be identified. But, as you also verified, Table 6 provided some public information about this topic.

The state of CLT industry development is closely associated with the time period in which CLT started to developed. Europe implemented CLT earlier than other countries, which explains the higher number of facilities there. The authors should address this aspect either in the review methodology or discussion.

R: We really appreciate your tips. We addressed this information together with some citations as you can verify in the second page of discussion. Thank you. :)

Reviewer 5 Report

The article includes a broad summary of glued CLT systems. I recommend updating the data on the production plants of CLT panels in the world. For example, in the Czech Republic there is not only one production plant (Novatop), but also others, such as Stora Enso or Nema. It would be advisable to check other relevant data to keep the article up to date.

1. What is the main question addressed by the research? - Comprehensive review to discuss cross-laminated timber (CLT). 2. Do you consider the topic original or relevant in the field? Does it address a specific gap in the field? - This is the state of the art in the field of CLT production and manufacturers. 3. What does it add to the subject area compared with other published material? - State of the art in common articles are usually focused on a narrow area of research that the articles expand on. This article discusses the state of the art broadly, but does not follow it up with further research. 4. What specific improvements should the authors consider regarding the methodology? What further controls should be considered? - Some data are not up-to-date (list of factories) and it would be advisable to supplement them. 5. Are the conclusions consistent with the evidence and arguments presented and do they address the main question posed? - yes, 6. Are the references appropriate? - Yes, they are. 7. Please include any additional comments on the tables and figures.

Author Response

The article includes a broad summary of glued CLT systems. I recommend updating the data on the production plants of CLTpanels in the world. For example, in the Czech Republic there is not only one production plant (Novatop), but also others, such asStora Enso or Nema. It would be advisable to check other relevant data to keep the article up to date.

R: We really appreciate your tips. It is worth mentioning that the map (Figure 5) shows the distribution of national companies according to their origins per nation, that is, the place of its headquarters. This point was duly emphasized in the figure title and figure mention. About the amount of factories, due to greater number, is described in the Appendix. For example, Stora Enso factory was already mentioned in the first version of our paper. In contrast, we verified that Nema was not inserted. We really appreciate your tip. As you can confirm, we revised again the list and, after exhaustive searches, a new list was developed and an update was done to insert new companies from Japan. Thus, we revised the list and we updated the observation, inserting these new companies in our evaluation, improving the sectorial sampling.

  1. What is the main question addressed by the research? -Comprehensive review to discuss cross-laminated timber (CLT).

R: We really appreciate your efforts in this reviewing process. Together with the comprehensive review, a representative sectorial survey was led too.

  1. Do you consider the topic original or relevant in the field? Does it address a specific gap in the field? - This is the state of the art in the field of CLT production and manufacturers.

R: Yes. After the systematic review, an unprecedented listing of CLT developers was obtained and a sectorial survey with this public was led to analyze the different aspects through a very representative sampling process.

  1. What does it add to the subject area compared with other published material? - State of the art in common articles are usually focused on a narrow area of research that the articles expand on. This article discusses the state of the art broadly, but does not follow it up with further research.

R: Thanks for your opinion. The following paragraph suggested a future way for nations: “In addition, the main attention should prioritize scientific research to identify the most appropriate silvicultural species for each territory and study the native species most suitable for conversion into engineered products. Even so, it is possible to consider some nations with the best future potential for the development of the production and market towards engineered wood products and, consequently, the installation of new factories and efficient manufacturing lines to produce modern construction inputs such as CLTs.”. Even so, we inserted further research in the conclusion section.

  1. What specific improvements should the authors consider regarding the methodology? What further controls should be considered? -Some data are not up-to-date (list of factories) and it would be advisable to supplement them.

R: Data was update. It is worth mentioning that we observed each individual according to its main origin, insofar as there are companies with two or more factories (some of them in different nations) and they usually provide a single website for the whole corporation. Thus, to avoid data duplication from a same corporation, every CLT developer with multiple manufacturing plants was specifically classified according to its main origin (Figure 5), that is, headquarters / head office location. However, those companies with multiple manufacturing plants were detailed in Appendix, whose tables listed all plants and locations. This consideration (to avoid data duplication) was also inserted in the initial part of sectorial survey results, as a strategy to detail our observations and considerations. In addition, the list of companies was updated and those companies prospected in this review stage were also inserted (and considered) in our evaluation, which is why the number of representativeness of sampling was improved.

  1. Are the conclusions consistent with the evidence and arguments presented and do they address the main question posed? - yes,

R: Thanks for your opinion.

  1. Are the references appropriate? - Yes, they are.

R: Thanks for your opinion. To satisfy other reviewer, references were expanded with the insertion of some recent contributions.

7. Please include any additional comments on the tables and figure.

R: Thanks for your opinion

Reviewer 6 Report

This work identifies the current state of the CLT industry over the world. The reviewer appreciates the efforts made for this work, and several questions or suggestions are listed below:

General questions:

1. Perhaps it is better to avoid using 'we' in the abstract.

2. The format of each section's subtitle should be consistent, either using capitals or not.

Specific questions:

1. It needs to state what goal this work serves, does it only provide information or could it strengthen collaborations? 

2. I can see China's research output is listed in the top 10 but why it has zero manufacturing plants?  

3. I think it might be a good point to relate the current industry with existing building codes related to CLT. As standards and policies have an influence on the development of this EWP. Could the authors provide more information on this?

4. It is important to link or identify the possible reasons behind the numbers provided. 

Author Response

This work identifies the current state of the CLT industry over the world. The reviewer appreciates the efforts made for this work, and several questions or suggestions are listed below:

R: We really appreciate your opinions and efforts in this reviewing process.

General questions:

  1. Perhaps it is better to avoid using 'we' in the abstract.

R: We corrected it.  Thank you.

  1. The format of each section's subtitle should be consistent, either using capitals or not.

R: Oh, it was our fault. Sorry. We corrected it. Thank you.

Specific questions:

  1. It needs to state what goal this work serves, does it only provide information or could it strengthen collaborations? 

R: we rewrote this part. Thank you. Both goals can be included from our findings and discussions. The issues were updated too.

  1. I can see China's research output is listed in the top 10 but why it has zero manufacturing plants?  

R: Thank you. Really it is slightly strange. But the inexistence of a wood culture in China to the detriment of a greater use of bamboo, due to extensive bamboo forests and low use of wood in construction in that nation, may justify this condition. We inserted a specific comment in the discussion to detail this remark. You can check the new comments in “In contrast, China do not have any industry plant dedicated to CLT (Figure 5), despite its extensive forests as measured by [43] and some internal market evinced by few websites with Chinese language (Figure 8a). In addition, this scenario may be incipient due to the bamboo culture in construction [47], limiting the use of wood products in that region.”.

  1. I think it might be a good point to relate the current industry with existing building codes related to CLT. As standards and policies have an influence on the development of this EWP. Could the authors provide more information on this?

R: Perfect, we provided the information related to standards in the discussions. In contrast, there is a gap on policies as confirmed by citation #1, which suggests the development of policies on CLT for housing.

  1. It is important to link or identify the possible reasons behind the numbers provided. 

R: Many reasons were commented behind the numbers provided. For example, a major concentration in Europe was explained in this revised version in view of the development of this panel in countries from this region, as well as the intensification of the wood utilization in these countries. The presence of some companies in other nations was explained by the reason of the existence of forest areas (native/planted) in those locations. Such facts were mentioned in the discussion.

Round 2

Reviewer 1 Report

Some references have been improved in the revision. Whatever the position of the reviewer about the virtual-based and web-search tools between a scientific publication and a benchmark study that can probably differ worldwide, the methodology is correctly performed and discussed. Table 6 and 7 completed by appendix are of interest in the timber community. Authors answer directly that this publication is a continuation of the previous study. To reviewer point of view doesn’t require a new publication and set in appendix on the first was enough.

Author Response

Some references have been improved in the revision.

R: Yes. We really improved this part in the last review. Thank you.

Whatever the position of the reviewer about the virtual-based and web-search tools between a scientific publication and a benchmark study that can probably differ worldwide, the methodology is correctly performed and discussed.

R: Our study can probably differ from other different regions, although any study will usually differ from each other due to their different perspectives, observations and, of course, distinct considerations. On the other hand, results from global views tend to reach similarities, above all, in high representations like ours. We appreciate your observations.

Table 6 and 7 completed by appendix are of interest in the timber community.

R: They are really unprecedented gifts for timber community, as tables 6/7 identify the future of the sector with the presentation of the present perspective of this global industry in the appendix, including contexts, available manufacturers, active factories, and respective geographic information. After publication, both forestry and construction suppliers and professionals will gain the access of the present sectorial scenario and the main active markets through the existing company concentrations.

Authors answer directly that this publication is a continuation of the previous study. To reviewer point of view doesn’t require a new publication and set in appendix on the first was enough.

R: We understood, but this information was required by other two reviewers as a route of hyperlinking two observations conducted on the same subject using different perspectives. While the first study consisted of a systematic study to address and discuss CLT products, the present manuscript is a sectorial analysis about the respective industry, being that they are part of the same postdoctoral research study led by the first author under the supervision of the last one. Thus, we will follow the suggestion of other reviewers on this point. In general, we appreciated your efforts to assist us in this process.

Reviewer 3 Report

Most of my previous concerns have been addressed in the revised manuscript. I believe the manuscript deserves to be published in the journal.

Author Response

Most of my previous concerns have been addressed in the revised manuscript. I believe the manuscript deserves to be published in the journal.

R: We really considered your remarks to improve our manuscript and discussions. We appreciated your efforts to assist us in this process.

Reviewer 6 Report

Thank you for your effort in finishing this manuscript. Please consider the following questions:

1. Could the authors address the problem in the CLT industry in the abstract? And perhaps why is it a problem? What purpose does the manuscript serve?

2. Regarding the development of CLT in China, it is more likely to be related to the perception of people, but there are many other reasons as well. For those living in cities with steel and RC structures, it is rather difficult to recognise timber structures as they are considered less reliable and more vulnerable to earthquakes and fires. The current research progress in China and the related codes are not complete and yet need to be improved. But this is just part of the reason. In addition, the bamboo industry and its application are rather new to the timber industry relating to modern structures. 

3. Could the authors reason why the UK does not have CLT developers? We have seen some CLT buildings there, as they used to be the tallest CLT structures in the world. The reviewer is interested to know the dominating factors for a country with many CLT projects but does not have domestic factories, which could, in the long term, lower the construction and transportation cost. 

4. Could the authors explain what problems this manuscript solves as a systematic review? 

5. Would the authors relate the status of the CLT industry in a country with the number of completed and ongoing CLT projects? 

Author Response

Thank you for your effort in finishing this manuscript. Please consider the following questions:

R: We really considered your remarks to improve our manuscript and discussions. We appreciated your efforts to assist us in this process.

  1. Could the authors address the problem in the CLT industry in the abstract? And perhaps why is it a problem? What purpose does the manuscript serve?

R: the existing “problem”, which is understood as the gap, was already cited by the abstract in the sentence “The current scenario of CLT developers was raised by that previous paper as one of the missing factors in the available literature, reason why this gap became the main goal of the present study.”. As the previous study raised this information, it is important to link both studies to justify this current demand previously observed. The reason of this problem is an evident gap on the industry information, as publications have been focused on product evaluation and development, which is why few industry analyses and remarks have been found in the literature. In addition, we inserted a new sentence to justify the purpose of our study, which was given by “A global perspective was led to provide information and discussion to all possible stakeholders.” Thank you for these remarks.

  1. Regarding the development of CLT in China, it is more likely to be related to the perception of people, but there are many other reasons as well. For those living in cities with steel and RC structures, it is rather difficult to recognise timber structures as they are considered less reliable and more vulnerable to earthquakes and fires. The current research progress in China and the related codes are not complete and yet need to be improved. But this is just part of the reason. In addition, the bamboo industry and its application are rather new to the timber industry relating to modern structures.

R: These perceptions likely exist not just in China but around the world, being verified by many studies on population and professional perceptions on the wood utilization in construction, whose causes are those cited by you, including possible wrong perceptions vulnerability and reliability. Even so, we re-wrote the sentence “Even with a recent code on the design of timber structures [47], this scenario is still incipient, as Chinese construction has been more traditionally related to a bamboo culture [48]” to insert a new citation to explain the existence of a code, although its culture is traditionally marked by bamboo.

  1. Could the authors reason why the UK does not have CLT developers? We have seen some CLT buildings there, as they used to be the tallest CLT structures in the world. The reviewer is interested to know the dominating factors for a country with many CLT projects but does not have domestic factories, which could, in the long term, lower the construction and transportation cost.

R: What? Of course UK has developers, which totalizes three companies!! While Ireland already presents two CLT producers (Cedarlam, and Glenfort Timber Engineering), England has another (Buckland Timber). You may confirm this information through Figure 5 and Table A2. Even so, in our globalized moment, CLT products are marketed to different nations, which is why there are products exported to other nations. Thus, the lack of companies is not a problem for leading nations, since there are effective housing / real estate investors interested in the use of CLT, which is not the case in the UK with its 3 local developers.

  1. Could the authors explain what problems this manuscript solves as a systematic review?

R: The manuscript does not “solve” problems only with a systematic review, as this systematic review is only the first part of a blended research formed by this methodology and completed by a sectorial survey with virtual-based analysis through website searches. In practice, any systematic review is used to verify the current perspective of any subject in the scientific literature. In our case, the results of this systematic review emphasized a visible lack of similar contributions on this global approach towards CLT industry.

  1. Would the authors relate the status of the CLT industry in a country with the number of completed and ongoing CLT projects?

R: This point is very important, although there is an effective lack on this kind of updated information. However, we inserted this demand as a recommended research line to be regarded by future studies in the final paragraph of the conclusion section.

Round 3

Reviewer 6 Report

After the revision, the manuscript is sufficient and clear on the topic. 

Regarding the industry in China, it is not appropriate to state that 'Chinese construction has been more traditionally related to a bamboo culture.' They have used timber and bamboo for ancient buildings, but timber is preferred over bamboo. We have visited some bamboo structures, but these are in rural areas for residential purposes, as well as they need to consider the available materials.  Please rewrite the sentence.

Author Response

REVIEWER 6

Regarding the industry in China, it is not appropriate to state that 'Chinese construction has been more traditionally related to a bamboo culture.' They have used timber and bamboo for ancient buildings, but timber is preferred over bamboo. We have visited some bamboo structures, but these are in rural areas for residential purposes, as well as they need to consider the available materials.  Please rewrite the sentence.

R: Okay. Despite the bamboo is intrinsically utilized by Chinese construction until the present times, this paragraph was slightly changed to suit your local experiences. You may confirm it through: “Yet, China has no CLT-focused factories (Figure 5), despite its extensive forests cited by [43] and its smaller market evinced by few websites with Chinese language (Figure 8a). Even with its recent code on the design of timber structures [47] and bamboo cultural heritage since ancient buildings [48], CLT is still incipient in China compared to lumber, steel and cement.”. We appreciated your efforts to assist us in this process.